# Single-cell transcriptomic analysis highlights origin and pathological process of human endometrioid endometrial carcinoma

Xiaojun Ren[1,2,5], Jianqing Liang[3,4,5], Yiming Zhang[3,4,5], Ning Jiang[3,4,5], Yuhui Xu[1,2], Mengdi Qiu[3,4], Yiqin Wang[1,2], Bing Zhao ●[3,4] ✉ & Xiaojun Chen ●[1,2] ✉

Endometrial cancers are complex ecosystems composed of cells with distinct phenotypes, genotypes, and epigenetic states. Current models do not adequately reflect oncogenic origin and pathological progression in patients. Here we use single-cell RNA sequencing to profile cells from normal endometrium, atypical endometrial hyperplasia, and endometrioid endometrial cancer (EEC), which altogether represent the step-by-step development of endometrial cancer. We find that EEC originates from endometrial epithelial cells but not stromal cells, and unciliated glandular epithelium is the source of EEC. We also identify LCN2 + /SAA1/2 + cells as a featured subpopulation of endometrial tumorigenesis. Finally, the stromal niche and immune environment changes during EEC progression are described. This study elucidates the evolution of cell populations in EEC development at single-cell resolution, which would provide a direction to facilitate EEC research and diagnosis.

Endometrial cancer (EC) is one of the most common gynecologic malignancies with increasing incidence worldwide[1]. As the major pathological type of EC, endometrioid endometrial cancer (EEC) accounts for 65–85% of all ECs[2,3]. Given the significance, a comprehensive understanding of EEC tumorigenesis would improve the diagnosis and treatment of the disease.

In estrogen-dependent EEC tumorigenesis, the endometrium exhibits uncontrolled proliferation under prolonged exposure of estrogen without progesterone protection and can develop from normal endometrium, to atypical endometrial hyperplasia (AEH, precancerous stage of EEC), then to EEC step-by-step[2,4,5]. Regarding which lineages EEC originates from, previous studies speculated that various lineages, including epithelial and stromal stem components of the endometrium, might be the origin of EEC[6–10]. Recently, a study reports that Axin2-positive epithelial stem cells in mouse endometrium could develop into EEC with the activation of Ctnnb1[11]. To date, the evidence is insufficient to support a clear origin of EEC.

Single-cell RNA sequencing (scRNA-seq) emerges as a powerful technique for evaluating the global gene expression profile of thousands of individual cells. These cells can be classified into specific clusters based on their characteristic transcriptomes at single-cell resolution. The analysis of cell population evolutions and the transcriptional changes in endometrial pathology, from normal endometrium to EEC, could help understand the pathological progression of EEC.

Tumor microenvironment (TME), which consists of immune cells, fibroblasts, pericytes, etc., plays an important role in tumorigenesis, prognosis, and metastasis[12–16]. Although previous studies have implicated the prognostic and potential therapeutic resistant role of TME in EEC[15,17], the landscape of TME from the normal endometrium to the formation of EEC remains elusive. Uncovering the characteristic changes of TME from normal endometrium to EEC would determine the functions of TME in EEC tumorigenesis.

In this work, we elaborate the single-cell transcriptome atlas of 15 endometrial tissues representing different stages of EEC development,

[1]Obstetrics and Gynecology Hospital of Fudan University, Shanghai, China. [2]Shanghai Key Laboratory of Female Reproductive Endocrine Related Diseases, Shanghai, China. [3]State Key Laboratory of Genetic Engineering, School of Life Sciences, Fudan University, Shanghai, China. [4]Zhongshan Hospital, Fudan University, Shanghai, China. [5]These authors contributed equally: Xiaojun Ren, Jianqing Liang, Yiming Zhang, Ning Jiang. ✉e-mail: bingzhao@fudan.edu.cn; chenxiaojun1366@fckyy.org.cn

including normal endometrium, AEH, and EEC. We propose that EEC originates from unciliated glandular epithelium and identify LCN2 + / SAA1/2+ cells as a featured subpopulation of endometrial tumorigenesis. Our findings provide valuable insights into the mechanism of EEC tumorigenesis.

## Results

### Cell population landscapes of normal endometrium, atypical endometrial hyperplasia, and endometrioid endometrial cancer

To elucidate the dynamic changes of cell components in endometrial tissues during the progression from normal endometrium to EEC, we profiled single-cell transcriptomes of endometrial tissues from 15 donors. These tissues represent the two different stages of EC development and they include five normal endometrial tissues, five atypical endometrial hyperplasia (AEH) tissues, and five EEC tissues (Supplemental Fig. 1 and Supplemental Table 1). Among the five EEC cases, three were classified as NSMP (no specific molecular profile) and two were classified as MMRd (mismatch repair deficient) (Supplemental Table 1). Transcriptomic analysis of these tissues was performed on a 10X Genomics system (Fig. 1a). For each tissue sample, we isolated single cells without prior selection for cell types. After quality filtering, a total of 99,215 cells were collected and yielded a median of 2317 detected genes per cell. The numbers of cells and genes from each tissue sample are provided in Table S1. Principal component analysis (PCA) showed that the normal endometrial samples were clustered together, while the AEH and EEC samples were scattered (Supplemental Fig. 2a), representing transcriptome differences in normal endometrial, AEH and EEC tissues.

We performed dimensionality reduction and unsupervised cell clustering using methods implemented in the Seurat software suite[18], followed by removing batch effects among multiple samples (see Methods section), and the results were presented using t-distribution stochastic neighbor embedding (t-SNE) (Supplemental Fig. 2b, c). Based on previously reported canonical cell markers, six main known cell types were identified (Fig. 1b, c): epithelial cells (*EPCAM, CDH1*)[11,19], stromal fibroblasts (*DCN, COL6A3*)[19], endothelial cells (*PECAM1, PCDH17*)[19], lymphocytes (*CCL5, STK17B*)[19], macrophages (*MS4A6A, CD68*)[19,20], and smooth muscle cells (*ACTA2, RGS5*)[19,21]. A recent study divides the epithelium into two cell types: the "ciliated epithelium" and the "unciliated epithelium" which may be responsible for different functions[19]. In our experiment, we also found these two distinct epithelial cell types: the ciliated epithelium (*EPCAM + CDHR3 + FOXJ1 +* ) and the unciliated epithelium (*EPCAM + CDHR3-FOXJ1-*) (Supplemental Fig. 2e). Expression heatmaps of the top 15 signature genes for each defined cell type reflected the specificity of cell population (Supplemental Fig. 2d).

### Increase of the epithelial proportions and copy number variants (CNVs) in endometrioid endometrial carcinoma

Increased epithelium is the pathological manifestation of endometrial carcinoma[22]. However, dynamic changes in the proportion of all cell types from normal endometrium, to AEH, and to EEC have not been reported. We analyzed the percentages of each cell type in different pathological stages. Muscle cells were not included in our analysis, as they are not in the endometrium. Evidently, the percentages of epithelial cells and stromal fibroblasts changed drastically during the pathological progression of EEC. The proportion of epithelial cells increased in AEH and further expanded in EEC, while the proportion of stromal fibroblasts dramatically decreased from normal endometrium to EEC (Fig. 1d). Changes in the proportions of epithelial cells and stromal fibroblasts are also verified by immunofluorescence co-staining of EPCAM (marker of epithelium) and VIM (marker of stroma) of histological sections (Fig. 1e). The proportion of endothelial cells did not show significant changes in different pathological stages during the development of EEC (Supplemental Fig. 2f). Although the percentage of

lymphocyte and macrophage was significantly increased in AEH compared with that in normal endometrium, the change was not significant in EEC, which may be related to the limitation of sample size (Supplemental Fig. 2f). Further investigation of cell population similarities showed that the epithelial cells in AEH and EEC greatly differ from that in normal tissues (Fig. 1f). Analysis of inferred copy number variants (CNVs) from different pathological stages indicated that epithelial cells in AEH and EEC have significant deviation from normal endometrium, which serves as a control (Fig. 1g). The distribution of CNVs in these five EEC samples did not show difference according to molecular subtypes (Supplemental Fig. 3a). Meanwhile, we validated in two independent normal endometrial samples by whole-exome sequencing, which displayed stabilized patterns of copy number profile in all chromosomes, implying that the normal endometrial tissue had low CNVs in cells and could be used as a baseline for AEH and EEC (Supplemental Fig. 3b). We next analyzed single cell CNVs of epithelial cells in AEH and EEC samples by inferring large-scale chromosomal CNVs based on transcriptomes using the normal epithelium as a control, and we found that high CNVs often occurs on chromosome 1, chromosome 8, or chromosome 10 (Supplemental Fig. 3c). This is consistent with the outcome of TCGA dataset[23], indicating that these may be canonical CNV subclones that could contribute to tumor progression. Nevertheless, the origin of these cells with high CNVs in cancer endometrium remains unclear.

### EEC originates from endometrial epithelial cells but not stromal cells

The hypothesis of EEC origin suggests that it may derive from mesenchymal stem cells or endometrial epithelial stem cells[6,9–11]. First, we tried to determine whether EEC cells originate in the stroma or epithelium.

Given that the increase in epithelial cell proportions is accompanied by the decrease of stromal fibroblast proportions, we posited that mesenchymal-epithelial transition (MET) might be involved in the endometrial pathological progression. We evaluated the global intrinsic characteristics of stromal fibroblasts. We found that stromal fibroblasts of endometrium from different pathological stages displayed high similarity to normal endometrium (Fig. 1f), and rare CNVs of stromal fibroblasts was observed in different pathological stages (Fig. 1g and Supplemental Fig. 3c). These results suggest that few transcriptome changes occurred in stromal fibroblasts. We then performed RNA velocity analysis to determine the relationship between epithelial cells and stromal fibroblasts for samples representing each pathological stage. As shown in Fig. 2a, RNA velocity predicted two independent trajectories in epithelial cells and stromal fibroblasts, indicating that a lineage trajectory of MET may not be valid. In addition, the expression of MET regulators *ELF3*, *OVOL1*, and *OVOL2*[24–26] were nearly not detectable in stromal fibroblasts (Fig. 2b and Supplemental Fig. 4). In conclusion, our data suggest that EEC most likely originates from endometrial epithelial cells, rather than stromal cells.

### Unciliated glandular epithelium is the source of EEC

Endometrial epithelium consists of glandular epithelium and nonglandular epithelium (also known as luminal epithelium), and both components contain ciliated and unciliated epithelial cells[19]. Therefore, having concluded that endometrial epithelial cells are the origin of EEC, we wanted to further investigate which kind of endometrial epithelium is the source of EEC.

To study the epithelial subpopulations of endometrial carcinoma, we used the FindClusters function and labeled 46 clusters according to distinct expression patterns by unsupervised clustering (Fig. 3a). In the split t-SNE map, we found that extra subpopulations (clusters 14 and 28) were present in AEH. Based on the cell clustering and the expression of signature genes (Fig. 1b and Supplemental Fig. 2e), we concluded that these were unciliated epithelial cells. Surprisingly, these subpopulations persisted in EEC sample (Fig. 3a), and this accords well

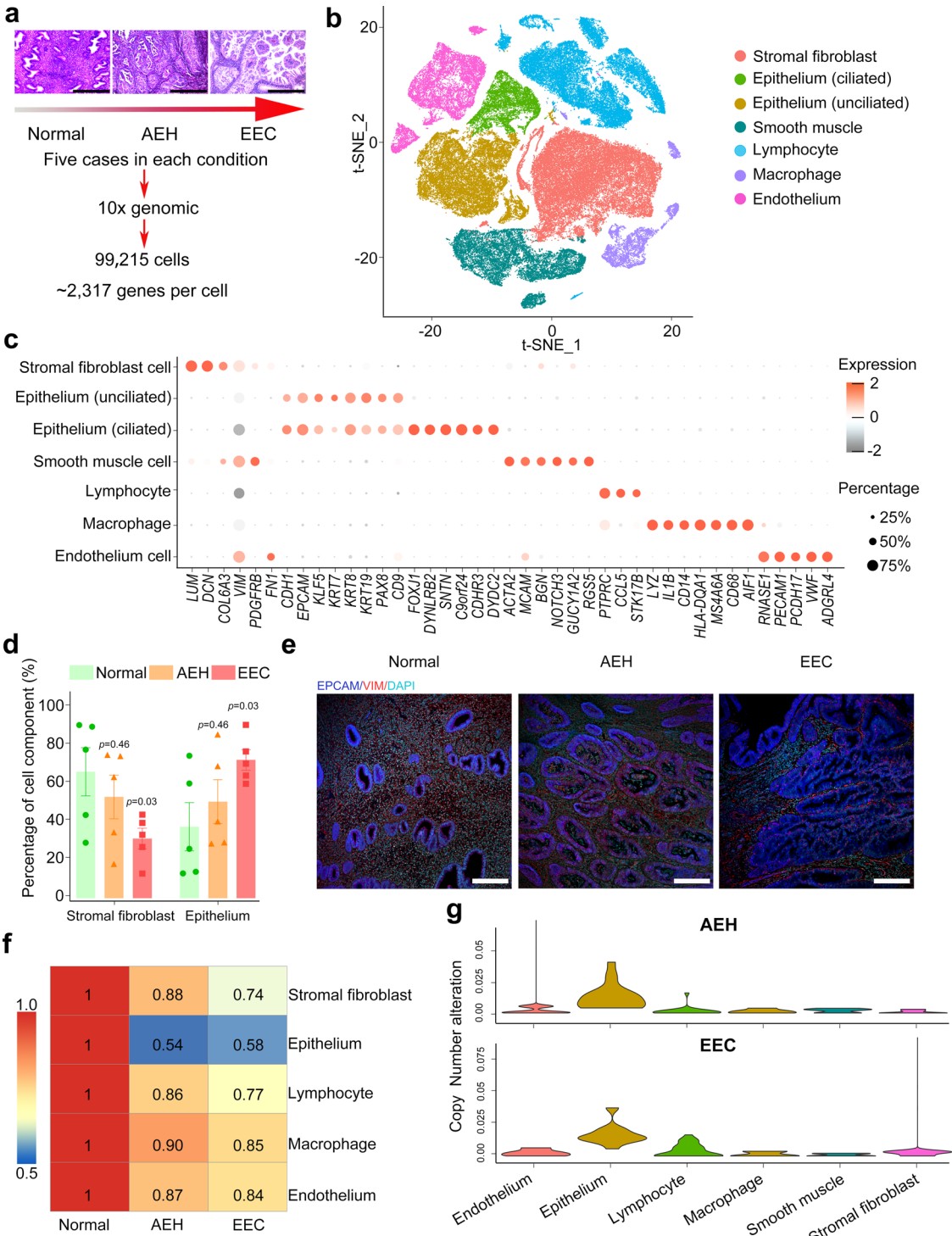

**Fig. 1 | Cell population landscapes of endometrial pathology. a** Overview of subjects enrolled in this study and representative H&E staining images were shown ($n = 5$ for each group, scale bar = 500 μm). Normal: normal endometrium; AEH: atypical endometrial hyperplasia; EEC: endometrioid endometrial cancer. **b** t-SNE plot showed six main cell type clusters of 15 specimens from different stages of endometrial pathology. **c** Expression patterns of canonical markers and differentially expressed genes (DEGs) of each cell type. Each dot represents a gene, of which the color saturation indicates the average expression level, and the size indicates the percentage of cells expressing the gene. **d** The percentage of stromal fibroblast and epithelial cells at different stages of endometrial pathology (Source data are provided as a Source Data file). Data are mean ± SEM based on 5 independent biological replicates. Significance was evaluated by comparing it with the normal group (t-test, two-sided; p values are denoted). **e** Epithelial and stromal cells of different endometrial pathology stages were identified by EPCAM and VIM via immunofluorescence ($n = 5$ for each group, scale bar = 100 μm). **f** Spearman correlation test was performed by comparing the expression of marker genes between different endometrial pathology stages (AEH and EEC) and normal sample. **g** Copy number variation score across different endometrial pathology stages (AEH and EEC) and their cell types.

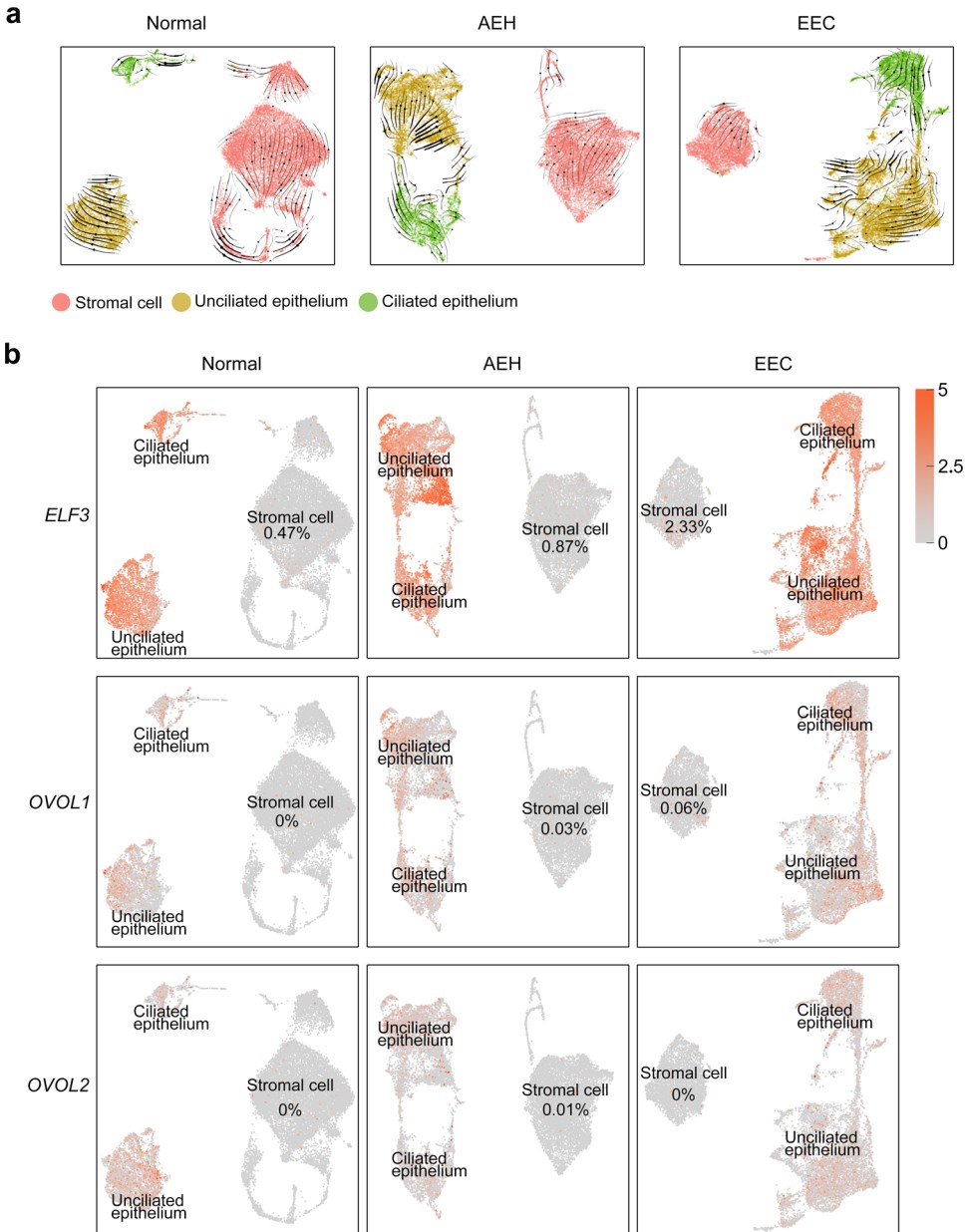

**Fig. 2 | Lineage trajectory of epithelium and stromal fibroblasts showed by RNA velocity analysis. a** RNA velocity suggests no lineage trajectory between epithelium cells and stromal fibroblast cells in different endometrial pathology stages. **b** Expression patterns of MET markers (*ELF3*, *OVOL1*, and *OVOL2*) were shown in stromal fibroblast and epithelial cells. The percentage represents the proportion of positive cells in stromal fibroblast (Source data are provided as a Source Data file).

with the observation that there was less similarity in epithelial cells and higher CNVs in AEH and EEC, when compared to normal endometrium. Additionally, a lineage trajectory from unciliated cells toward ciliated cells was identified in AEH and EEC samples (Fig. 2a), hinting that unciliated epithelium itself is more likely to be the origin of these emerging subpopulations than ciliated epithelium. Based on these results, we speculated that clusters (14 and 28) arising in AEH and EEC were the signal of the development of EEC, and most probably derived from the normal unciliated epithelium.

To further investigate the subpopulations of unciliated epithelium, we selected and re-clustered unciliated epithelial cells (*EPCAM + / FOXJ1-*) from normal endometrium and EEC samples, yielding 36 clusters (Supplemental Fig. 5a–c). Most clusters consisted of cells from multiple samples, indicating an unbiased cell subpopulation representation. Moreover, almost all cells from normal samples could be classified as glandular cells (*FOXA2, SMAD9*) or luminal cells (*WNT7A*) (Fig. 3b, c and Supplemental Fig. 5a), according to the reported

markers[19,27,28]. However, the EEC samples showed heterogeneity of cell clusters (Supplemental Fig. 5a). Aside from glandular and luminal cell groups, we also found that clusters 14, 15, 26, and 32 were commonly enriched in EEC samples (Supplemental Fig. 5a–d). Differentially expressed genes of these clusters showed distinctive related pathways, compared to other clusters: clusters 14 and 15 were mainly enriched in TNF and IL-17 signaling pathways, while clusters 26 and 32 were enriched in Hippo and Wnt signaling pathways that are related to cancer progression (Supplemental Fig. 5e), suggesting tumor heterogeneity and their co-regulation of tumor progression.

For clusters 14, 15, 26, and 32 exclusively appear in all EEC samples when compared with normal subjects, it indicates a commonality of inherent variation among the different EEC samples. Therefore, we considered them as oncogenic subpopulation (Fig. 3b). The heatmap showed the top 25 signature genes of the oncogenic subpopulation (Supplemental Fig. 5f), such as *LCN2, SAA1*, and *CFB* (Fig. 3c), representing the special characteristics of this subpopulation. Because cells

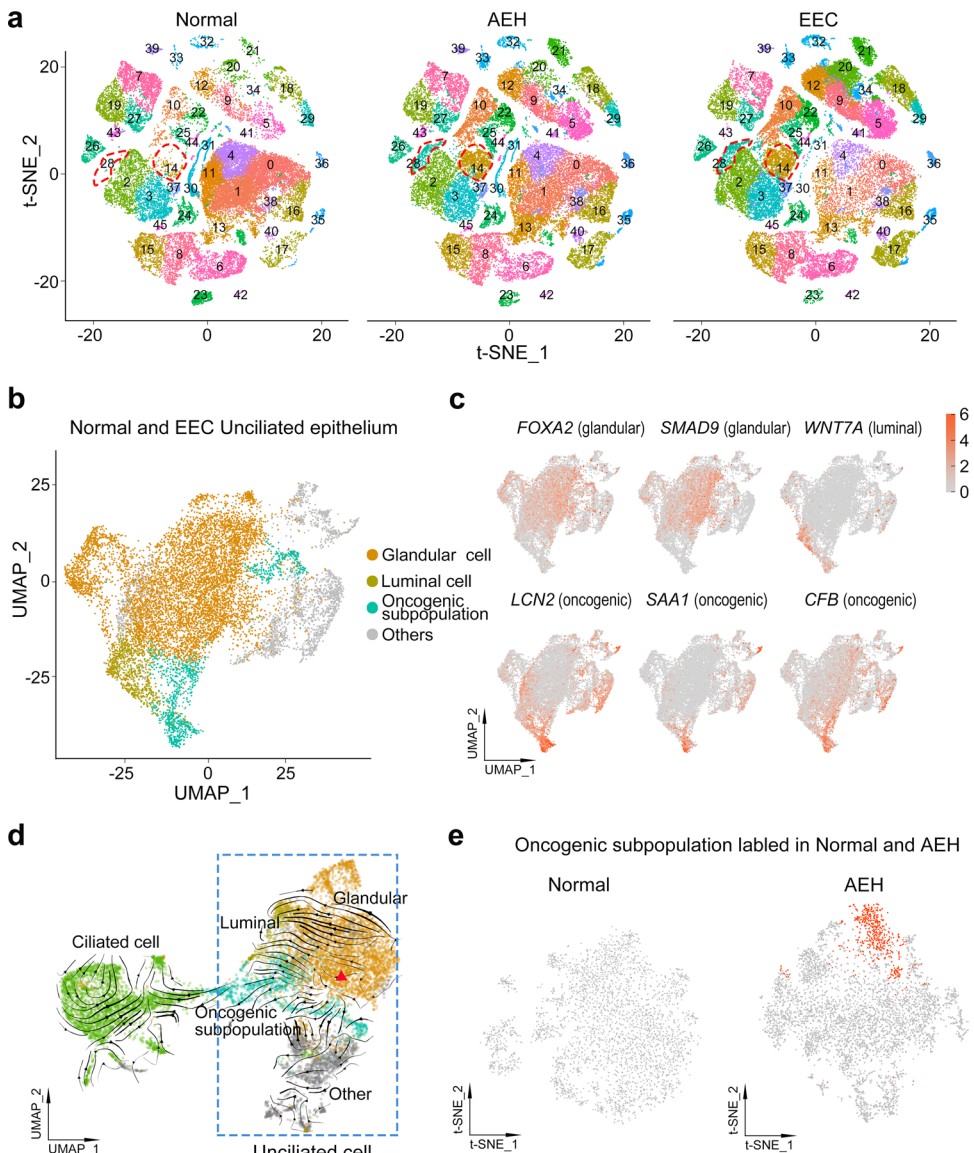

**Fig. 3 | Unciliated glandular epithelium is the source of endometrial carcinoma.**
**a** Split t-SNE plots of Fig. 1b for detail clusters show that clusters 14 and 28 are arising during tumor progression. **b**, **c** t-SNE plot of unciliated epithelium cell of normal and EEC showing distribution (**b**) and marker genes pattern (**c**) of glandular cell, luminal cell, and oncogenic subpopulation. **d** RNA velocity upon UMAP plot of epithelial cells of normal and EEC suggests that oncogenic subpopulation is generated from unciliated glandular cell. The red triangle indicates the initial position of RNA stream. **e** Projection of 'oncogenic subpopulation' on the t-SNE plot of normal and AEH unciliated epithelial cells, carried out by the TransferData function developed by Seurat team.

of oncogenic subpopulation are different from normal endometrium-derived epithelial cells, we infer that emergence of an oncogenic subpopulation is related to the progression of EEC.

To determine the origin of cells of the oncogenic subpopulation, we applied RNA velocity analysis to predict the future state of individual cells on a timescale to track the evolution trajectory of EEC epithelium. Notably, our velocity estimation showed a strong directional flow from a group of unciliated glandular cells to the oncogenic subpopulation (Fig. 3d), indicating unciliated glandular cells were most likely to be the source of cells exiting in the oncogenic subpopulation. It's well documented that AEH is the precursor lesion of EEC. To address whether an oncogenic subpopulation has been appeared in AEH, we re-clustered the unciliated epithelium groups from AEH samples and used the TransferData function to capture the oncogenic subpopulation defined in EEC samples (Fig. 3e). We found that cells with characteristics of oncogenic subpopulation were already present in AEH samples, but not in normal samples, with high expression of the same signature genes, including *LCN2* and *SAA1* (Fig. 3e and

Supplemental Fig. 6a). Moreover, the trajectory of epithelium in AEH displayed unciliated glandular cells were the putative cellular origin of oncogenic subpopulation as well (Supplemental Fig. 6b). These results indicate that the emergence of distinctive unciliated epithelium subpopulation is a critical step of endometrial cancerous progression and could be valuable in EEC early diagnosis.

## Identification of LCN2+/SAA1/2+cells as a featured subpopulation of endometrial tumorigenesis
To investigate the regularity of the oncogenic subpopulation signature genes in the pathological process of human endometrial carcinoma, we compared expression levels of the top 25 marker genes (Supplemental Fig. 5f) in unciliated epithelium among normal endometrium, AEH, and EEC samples. We found 10 signature genes (*S100A9, DKK4, S100A8, LCN2, CTS1, LTF, CXCL1, NOTUM, SAA1*, and *SAA2*), which exhibited significantly upregulated expression in EEC (Fig. 4a). The upregulation of *S100A9, S100A8, LCN2, LTF, CXCL1, SAA1*, and *SAA2* started at the AEH stage, indicating that these seven signature genes

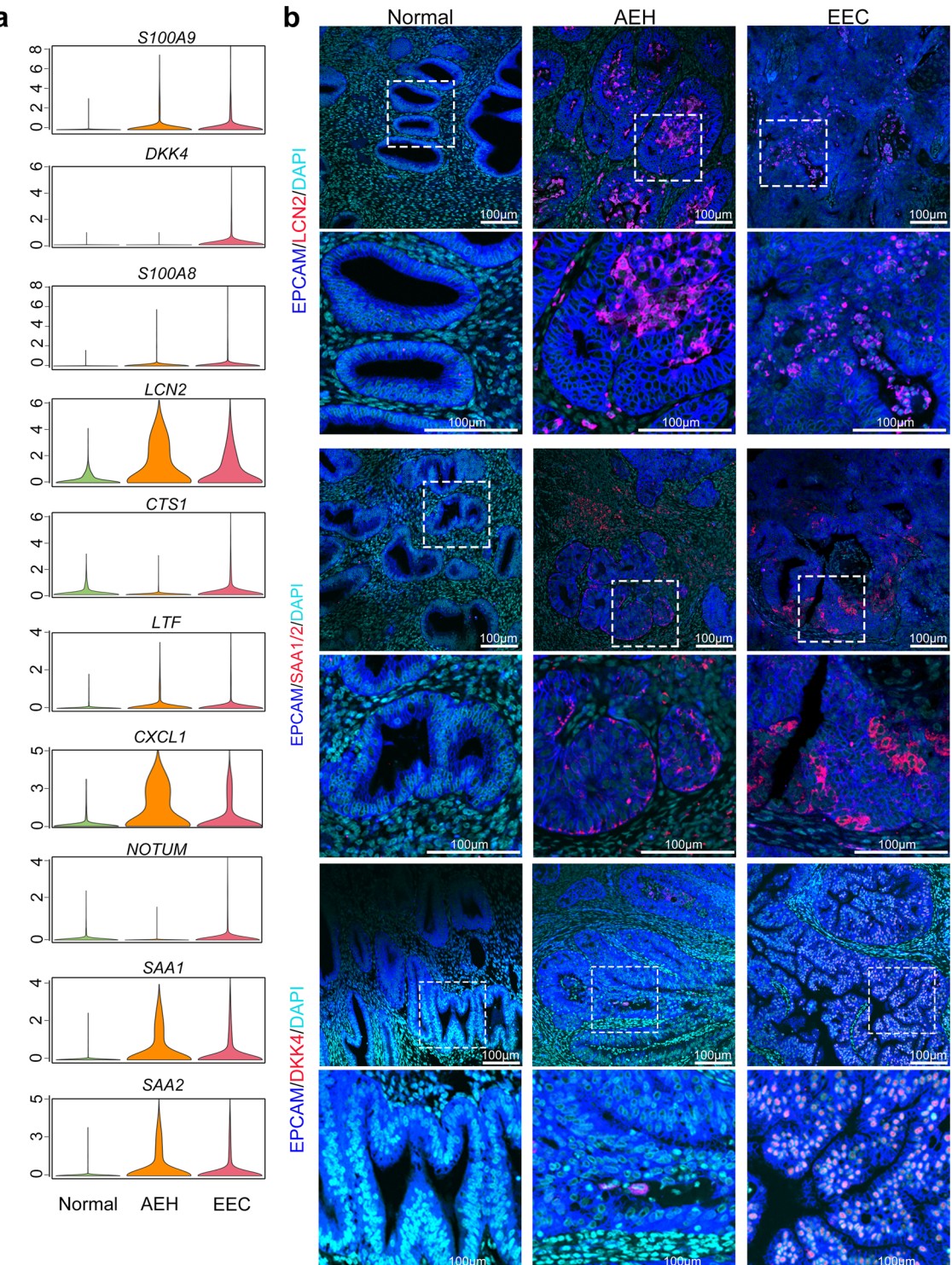

**Fig. 4 | Identification of oncogenic subpopulation on clinical specimens.**
**a** Violin plots showing mean expression of genes which are screened between EEC and normal samples, in different endometrial pathology stages (Normal, AEH and EEC). Candidate genes were chosen according to the third quantile of normalized expression in EEC and normal group holding a difference value larger than 0.9.
**b** Immunofluorescence was performed to identify LCN2, SAA1/2, and DKK4 positive cells, respectively, in different endometrial pathology stages (Normal, AEH and EEC; *n* = 3 for each group). Scale bar = 100 μm.

might be involved in the early events of tumorigenesis of EEC and could be used as the markers for early diagnosis of EEC. Additionally, immunofluorescent staining in different stage of endometrial lesions verified that EPCAM + /LCN2 + cells arose in AEH and EEC samples, but not in normal endometrium (Fig. 4b), which is consistent with the previous reports that LCN2 is over-expressed in cancers of diverse histological origin[29,30]. Similarly, it has been reported that SAA (serum

amyloid A) production is related to tumor tissues, such as colorectal, ovarian, uterine, glioblastoma cancers[31], and EEC[32]. However, whether SAA could serve as markers of early diagnosis of EEC remaining unclear. Leveraging to single-cell sequence, we found the expression of *SAA1* and *SAA2* (another acute-phase protein) increased in both AEH and EEC unciliated epithelium (Fig. 4a). Furthermore, immuno-fluorescent staining showed that SAA1/2 expression also appeared in

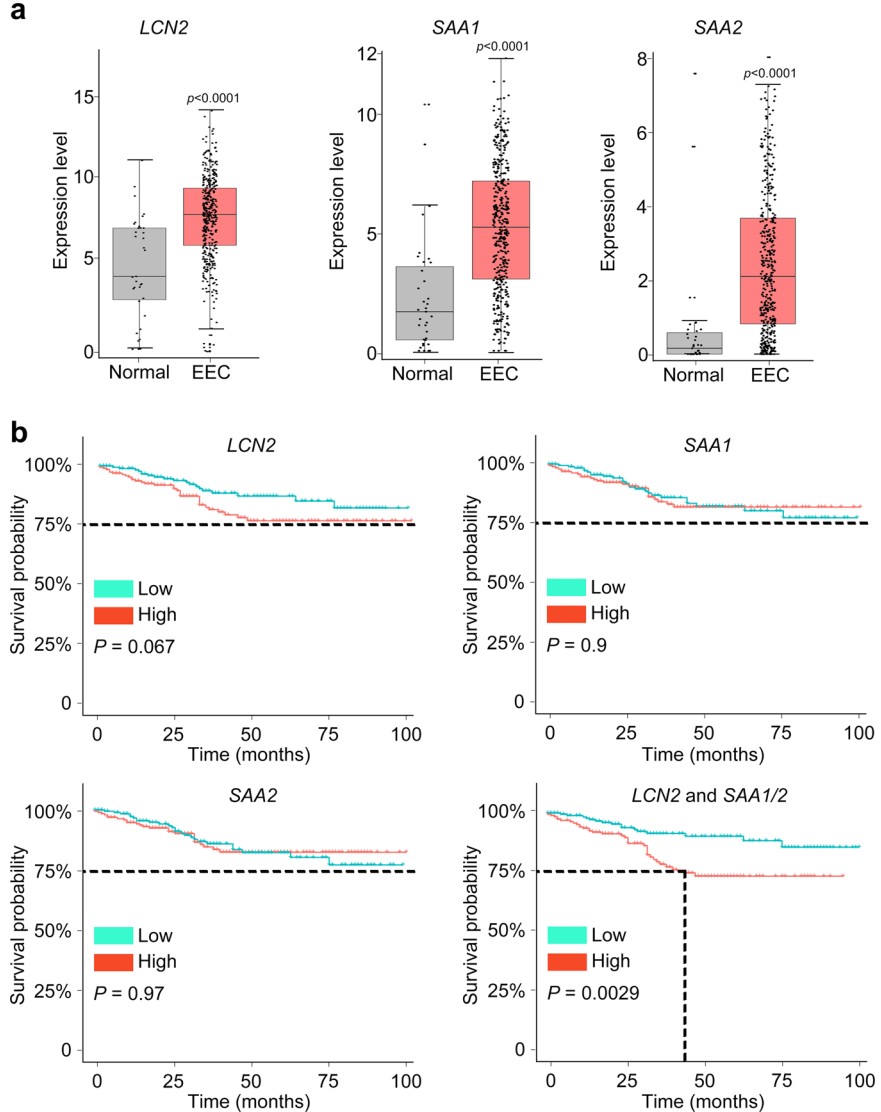

**Fig. 5 | Oncogenic subpopulation has implications for EEC patients' survival.**
**a** According to the TCGA EEC data, the expression of *LCN2*, *SAA1*, and *SAA2* was significantly elevated in EEC subjects (*t* test, two-sided, *p* values are noted). Normal samples: *n* = 35, EEC samples: *n* = 411. The expression level of each gene is visualized by log₂(TPM + 1). Bar plots show the median (center), 25–75 percentile (box), and lower whisker = smallest observation greater than or equal to lower hinge−1.5 * IQR

(interquartile range); upper whisker = largest observation less than or equal to upper hinge + 1.5 * IQR (Source data are provided as a Source Data file).
**b** Combination with LCN2 and SAA1/2 predicted a disease outcome across TCGA EEC cohorts. The median was set as the cut-off value to stratify EEC patients into low-risk and high-risk groups. The endpoint was 100 months. Survival curves were compared via Log−rank test and visualized by Kaplan–Meier method.

some epithelial cells of AEH and EEC samples, but not in normal endometrium, which is similar to the pattern of LCN2 (Fig. 4b). Altogether, we proposed LCN2 and SAA1/2 could serve as a feature of early endometrial tumorigenesis.

### Database analysis pinpoints the key signature genes of endometrial oncogenic progress

To further confirm the correlation between *LCN2* and *SAA1/2* expression and endometrium cancer, we analyzed The Cancer Genome Atlas (TCGA) cohorts and observed significantly upregulated *LCN2* and *SAA1/2* expression in EEC cases compared with normal samples (Fig. 5a). Patients with high-risk score which was calculated for the *LCN2* and *SAA1/2* signatures in endometrial lesion had significantly lower survival rate across TCGA EEC cohorts (Fig. 5b), suggesting that we identified a common subpopulation underlying high-risk subtypes.

We also found some candidate genes specifically expressed in EEC, such as *DKK4*, *CST1*, and *NOTUM* that are highly expressed in EEC but rarely in AEH (Fig. 4a, b). In conclusion, our study demonstrated

that an intrinsic LCN2 + /SAA1/2+ epithelial cell subpopulation serves as an early marker of EEC, and the development of *DKK4*, *CST1* and *NOTUM* expression may be a warning sign of the final cancerous stage.

### Stromal fibroblasts might be indispensable in both normal endometrium and EEC

In addition to being driven by its intrinsic changes, the progression of tumors is also affected by the tumor microenvironment (TME). Previous studies suggest that normal stromal cells may be induced into cancer-associated stromal fibroblasts (CAFs) to promote tumor growth[17,33]. Therefore, we characterized the stromal fibroblast in tumorigenesis to determine how stromal fibroblasts converted in EEC and how they influenced the progression of tumor.

Unlike in epithelial cells, the CNVs and heterogeneity of cell clusters in stromal fibroblasts showed insignificant transcriptome variations among normal, AEH, and EEC (Fig. 1f, g and Supplemental Fig. 3c). And t-SNE map showed well overlap of stromal fibroblast single-cells between normal endometrium and EEC (Fig. 6a and

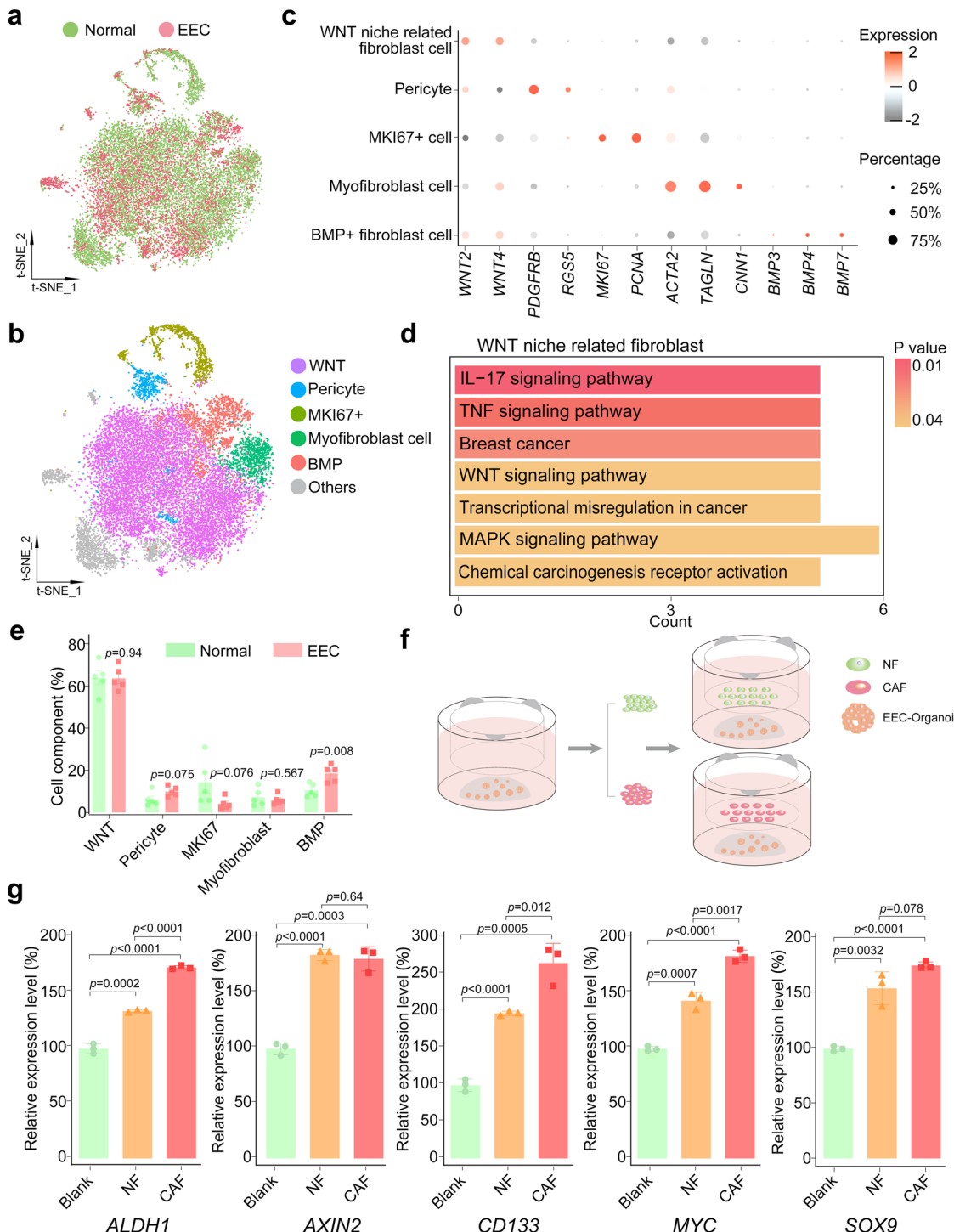

**Fig. 6 | Converted stromal fibroblasts establish oncogenic signal niche for endometrial epithelium. a** t-SNE plot of stromal fibroblast from normal and EEC samples. **b** t-SNE plots of indicated cell-type clusters of stromal fibroblast cells. **c** Expression patterns of canonical markers of fibroblast cell types. Each dot represents a gene, of which the color saturation indicates the average expression level, and the size indicates the percentage of cells expressing the gene. **d** KEGG enrichment analysis of differential expression genes between WNT niche-related fibroblast cells and others was enriched in Wnt signaling pathway. *P*-values are analyzed by Fisher's exact test, one-sided. **e** The percentage of different cell-type of stromal fibroblast showing in (**b**) between normal and EEC samples. Data are mean ± SEM based on five independent biological replicates (Source data are provided as a Source Data file). *P*-value was analyzed by *t*-test, two-sided. P values

are denoted above. **f** Co-culture scheme of EEC organoids and fibroblasts. EEC organoids were seeded in the lower well, and fibroblast cells (obtained from normal endometrium or endometrial cancer tissues with EEC) were seeded in the upper well. After one week of co-culture, the total RNA of EEC organoids was collected for subsequent experiments. **g** Relative mRNA expression of the stemness-related genes (*ALDH1, AXIN2, CD133, MYC,* and *SOX9*) were confirmed by qPCR in EEC organoids which were co-cultured with normal endometrial fibroblast (NF) or cancer-associated endometrial fibroblast (CAF). Separate EEC organoids culture was used as a blank control (Blank). Data are represented as mean ± SD (*n* = 3); NF and CAF of the co-culture system were derived from two independent patients. *P*-value was analyzed by *t*-test, two-sided. *P* values are denoted above.

Supplemental Fig. 7a). The composition of the stromal environment is defined by expression of cell-associated markers (Fig. 6b). Generally, based on expression of marker genes, we identified pericyte (positive for *PDGFRB* and *RGS5*), myofibroblast (positive for *ACTA2* and *TAGLN*), and *MKI67* + cell populations (Fig. 6c). FindMarker function is used to further define other populations, and we defined WNT niche related fibroblast (positive for *WNT2* and *WNT4*) and *BMP* + fibroblast (positive for *BMP3*, *BMP4* and *BMP7*) (Fig. 6b, c) groups. Although *WNT2* and *WNT4* did not show exclusive enrichment by visualization of tSNE map (Supplemental Fig. 7b), KEGG analysis showed that the WNT signaling pathway was enriched in WNT niche-related fibroblasts (Fig. 6d), confirming our cluster definition. The cell type composition analysis manifested that the proportion of BMP subtype of stromal cells increased slightly, while that of WNT subtype, pericyte, and myofibroblast did not change significantly (Fig. 6e).

Therefore, we further examined whether stromal fibroblasts could serve the same function in normal endometrium and EEC. It had been reported that cancer-associated stromal fibroblasts (CAFs) exhibited a promotion effect in the progression of EC[17]. In order to confirm the importance of stromal fibroblast niche for the growth of epithelial tumor cells and to study the differences between stromal fibroblasts from normal endometrium and CAFs from EEC, we performed co-culture assay using EEC organoid (Fig. 6f). Relative expression level showed that stromal fibroblasts from both normal and cancer donors promoted upregulation of stemness-related genes (*ALDH1*, *AXIN2*, *CD133*, *MYC*, and *SOX9*) in EEC organoids (Fig. 6g). Compared with those from normal endometrium, stromal fibroblasts from EEC manifest with a higher expression of *ALDH1*, *CD133*, and *MYC*, implying that the CAFs in EEC has enhanced ability to support the proliferation of endometrial epithelium. Therefore, our data suggest that all subtypes of stromal fibroblasts might be indispensable in both normal endometrium and EEC.

### Disruption of immune environment permits endometrial tumorigenesis

Tumor-associated immune cells are also critical for the establishment of TME. As we aimed to identify proportion changes in immune subtypes of endometrial lesions compared to normal tissue, we selected all immune cells including lymphocytes and macrophages from normal, AEH, and EEC samples. Cell identity was assigned based on the expression of canonical markers genes[34,35], including major populations of *CD3D* + T cells, *CD68* + macrophages, *NCAM1* + NK cells, *CD1A* + dendritic cells, *CD79A* + B cells, *KIT* + mast cells, and *FCGR3B* + neutrophils (Fig. 7a and Supplemental Fig. 7c). We find that there is no significant change in the percentages of these major subtypes to the total immune cells at different pathological stages (Fig. 7b and Supplemental Fig. 7d).

To further deconstruct macrophages and lymphocytes, which account for a large proportion of immune cells, we analyzed their subcellular clusters respectively. We divided the macrophages into 5 distinct populations (Supplemental Fig. 7e). Based on the population definition of the classical macrophage model[36,37], we found that both M1 (positive for *IL1A* and *IL1B*) and M2 (positive for *CD163* and *IL10*) associated genes were frequently expressed in the same populations (Supplemental Fig. 7f). This result challenges the model of macrophage polarization (M1-type and M2-type). We tried to identify marker genes for each of these clusters and assign them to *FCN1* + (cluster1), *SPP1* + (clusters 0, 3, 4) and cycling (cluster 2, positive for *TYMS* and *MKI67*) macrophage (Fig. 7c, d). And there was no significant change in the proportion of these subtypes in macrophages between normal, AEH, and EEC samples (Fig. 7e). Recent studies show that *FCN1* + monocyte-like cells could give rise to *SPP1* + populations in colon cancer[38], and three distinct macrophage subsets marked by expression of *FCN1*, *SPP1*, and *FABP4* have been identified in human lung[39]. These indicate that *FCN1* and *SPP1* may be universal markers of macrophage.

GO analysis of these two distinct macrophage subtypes showed that *FCN1* + macrophages were related to positive regulation of cytokine production and cellular response, while *SPP1* + macrophages were related to positive regulation of lymphocyte activation (Fig. 7f). As expected, cycling subtype of macrophages was enriched in cell cycle-related pathways (Fig. 7f).

Moreover, detailed clustering of T lymphocytes is shown in Fig. 7g. Based on canonical marker genes[34], we annotated both *CD8* + and *CD4* + T cells. In order to further define the state of CD8 T cell and CD4 T cell, we classified CD8 T cells as naïve (positive for *TCF7*, *IL7R*, and *CCR7*), cytotoxic (positive for *GNLY*, *GZMA*, *GZMB*, and *GZMH*) and exhausted (positive for *CTLA4*, *PDCD1*, and *HAVCR2*), and CD4 T cells as naïve (positive for *TCF7*, *IL7R*, and *CCR7*), exhausted (positive for *CTLA4*, *PDCD1*, and *HAVCR2*) and regulatory (Treg cell, positive for *FOXP3* and *TNFRSF4*), based on the expression of reported marker genes (Fig. 7h and Supplemental Fig. 8). To determine the composition of T cells in the immune microenvironment of normal endometrium, AEH, and EEC, we evaluated the proportion of T lymphocyte subpopulations. Notably, the proportion of cytotoxic CD8 T cell is significantly reduced in EEC compared to normal samples, as did naive CD8 T cell (Fig. 7i). On the contrary, *FOXP3* + CD4 Treg lymphocytes, which is related to immunosuppression, is preferentially enriched in EEC samples (Fig. 7i), suggesting the potential mechanism of immune escape in EEC. Our results provide evidence for the differential distribution of CD4 and CD8 T cells in endometrium carcinoma immune microenvironment, and the increase of Treg and reduction of cytotoxic and naive cells may be characteristics of EEC.

## Discussion

Here we present a comprehensive comparison of the transcriptional characteristics of different types of endometrial cells during the development of EEC at single-cell resolution. Our study describes the dynamic changes of the endometrial lesion, and our results not only confirm many important observations that are previously reported but also provide information for a better understanding of EEC development. By identifying distinct cell subpopulations, delineating a lineage trajectory, and cataloging tumor microenvironment cell types, our dataset will fuel future advances in the basic research of EEC origin and progression.

Our data demonstrated dramatic epithelial and stromal component change as well as transcriptome variations in epithelium during the process of EEC. The endometrium is composed of multiple cell types that undergo various forms of differentiation and remodel at a rapid rate during each menstrual cycle to maintain tissue homeostasis. Once endometrial homeostasis is disrupted, lesions occur. Physiologically, a recent study has profiled the heterogeneity of endometrial cells at the levels of types, states, proliferation, and differentiation across the human menstrual cycle[19]. However, from the pathological viewpoint, the cellular heterogeneity in endometrial lesions have not been reported, which seriously limits our understanding of human endometrial transformation across EEC development. In addition to the general histological changes during the pathological progression from normal endometrium to EEC[22], we profiled dynamic changes of different cell components at single-cell resolution, confirming the representative characteristics of increasing epithelium and decreasing stromal fibroblasts in EEC, when compared with normal endometrium. Through the categorization of cell types, based on their correlation and CNVs analysis, we found dramatic transcriptome variations in epithelium but not stromal or immune cells, during the development of EEC. These findings are in accordant with the general understanding of EEC as epithelial cancer.

Our data support that EEC might originate from unciliated glandular epithelial cells. One popular assumption for the origin of endometrial cancer points to MET; the supporting evidence being that mitotic activity was observed in stromal but not epithelial cells, either

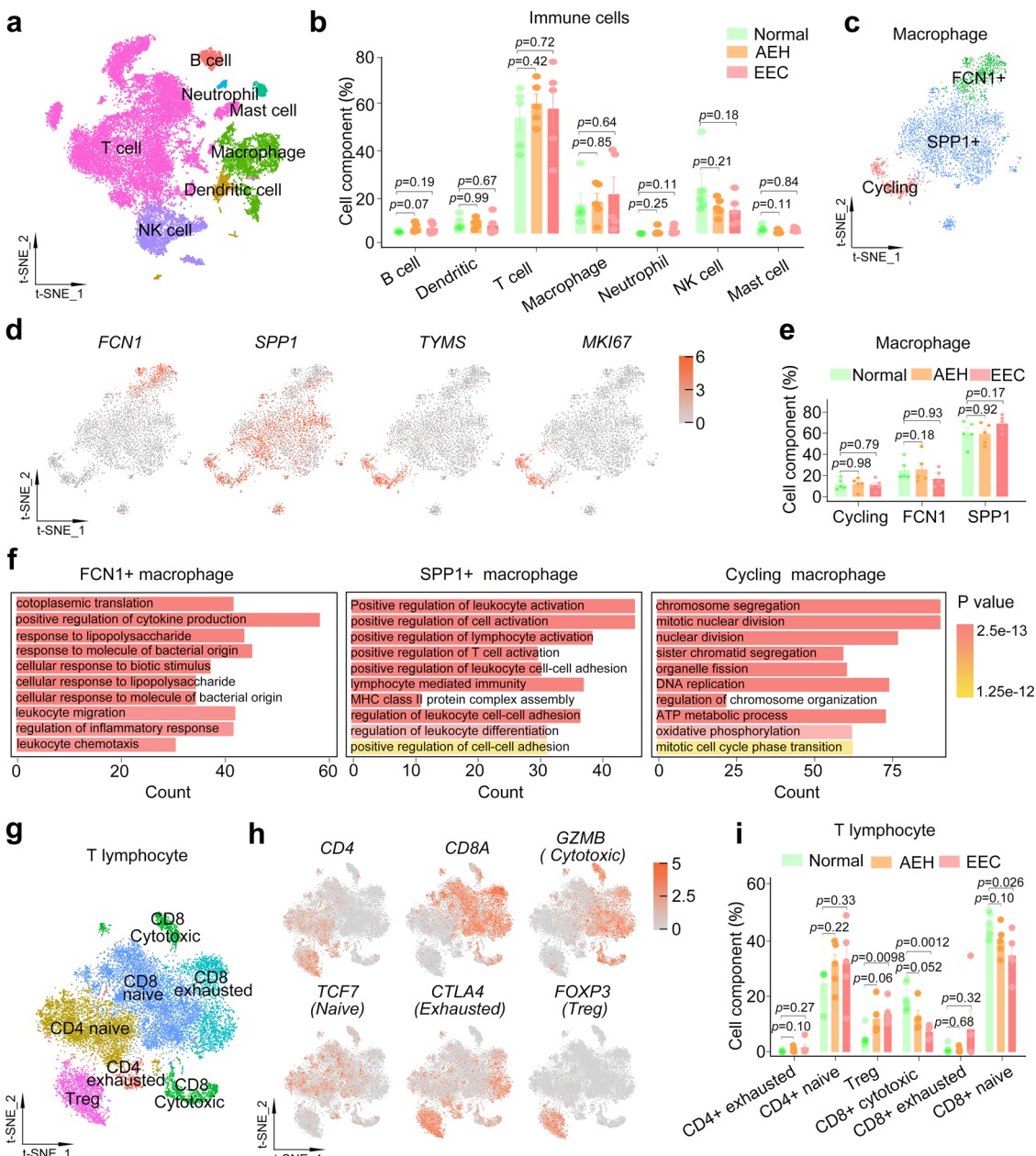

**Fig. 7 | Disruption of immune environment from normal to EEC. a** t-SNE plot of different cell types of immune cells from normal, AEH and EEC. **b** The proportion of different immune cell types at different stages of endometrial pathology. Data are mean ± SEM based on five independent biological replicates. *P*-value was analyzed by comparing with AEH or EEC versus normal group and there was no significance (*t* test for each group, two-sided, and *p*-values are denoted above). **c** t-SNE plot showing macrophage subtypes. **d** The expression of the signature gene of different macrophage subtypes. **e** The proportion of different subtypes in macrophage at different stages of endometrial pathology. Data are mean ± SEM based on five independent biological replicates. *P*-value was analyzed by comparing with AEH or EEC versus normal group and there was no significance (*t* test for each group, two-sided, and *p*-values are denoted above). **f** GO enrichment analysis of differential genes in each macrophage subtype. *P*-values are analyzed by Fisher's exact test, one-sided. **g** t-SNE plot showing T lymphocyte subtypes. **h** The expression of signature genes of T lymphocyte subtypes. **i** The proportion of different T lymphocytes subtypes at different stages of endometrial pathology. Data are mean ± SEM based on five independent biological replicates. *P*-value was analyzed by comparing with AEH or EEC versus normal group (*t* test for each group, two-sided, and *p*-values are denoted above). Source data are provided as a Source Data file.

in human[40] or mice[41,42]. However, the MET hypothesis for the origin of endometrial cancer is controversial. Ghosh et al.[43] performed lineage tracing in adult mice and found that mesenchymal cells could not give rise to epithelial cells in oviduct or uterus[43]. In our study, we performed RNA velocity analysis in different pathological stages and demonstrated that EEC originates from unciliated glandular epithelial cells but not stromal cells, providing supportive evidence for origin of EEC. Nonetheless, our observation needs to be confirmed by further lineage tracing in vivo.

We identified unciliated epithelial subpopulations (oncogenic subpopulation) through high resolution scRNA-seq, which arose in the endometrium, starting from the stage of AEH and persisted in EEC. By comparing the unciliated epithelial cells in EEC against those in normal endometrium, we captured EEC common groups and their enrichment pathways (TNF, IL-17, Hippo, and Wnt signaling pathways). Among these pathways, some have been reported to be associated with human endometrial disease. For instance, TNFα signaling pathway predicts poor prognosis of patients with endometrial cancer[44], IL-17 is

significantly higher in patients with chronic endometritis[45], Hippo/YAP-signaling pathway plays a critical role in the pathogenesis of endometriosis and endometrial fibrosis[46,47], and constitutive activation of Wnt/β-catenin signaling in the endometrium induces endometrial hyperplasia[48,49]. Studies have shown that these pathways are strongly associated with endometrial disease and even EEC progression, and therefore the emerging of extra unciliated epithelial subpopulations could be a powerful feature of EEC.

By further investigating the key signature genes of oncogenic subpopulation, we found that *LCN2* combined with *SAA1/2* can better predict the disease outcome. High level of *LCN2* is associated with increased cell proliferation, cell invasion, and metastasis[50], and also with aggressive features in EEC[51,52]. Meanwhile, *SAA* has been reported as a risk factor for several types of cancer[53–56]. But the behavior of *SAA1/2* in endometrial tumorigenesis has not been well understood. We found that the expression of *SAA1/2* increased in AEH and persisted in EEC, indicating that up-regulation of *SAA1/2* expression could be a sign of precancerosis of EEC. Analysis of the prognosis of EEC patients from TCGA-EEC data base also confirmed that the high-risk score calculating for *LCN2* and *SAA1/2* was associated with poorer overall survival, indicating the potential usage of *LCN2* and *SAA1/2* as biomarkers for early diagnosis of EEC. Taken together, we are inclined to conclude that EEC originates from epithelial cells, and the emergence of the oncogenic subpopulation could be an important indicator for EEC progression. Nevertheless, it is yet to be determined whether these genes are the main driving force of these subpopulations.

It is unclear how microenvironment (including the stromal and immune cells) is involved in endometrial carcinoma progression, especially during the step-by-step transformation from normal, to AEH, and to EEC. Our results agree with the widely known phenomena that the proportion of endometrial stromal cells decreases during the progression from normal endometrium to AEH and even lower in EEC. Although the relative number of stromal fibroblasts in EEC decreased, in vitro co-culture study showed that stromal cells from both normal endometrium and EEC could provide growth support for the EEC organoids. Particularly, CAFs showed a stronger up-regulating effect on the expression of stemness genes in EEC organoids, suggesting that the supportive effect of stromal fibroblasts could be enhanced in tumor microenvironment. Our results are consistent with the previous report, which concludes that CAFs promote the proliferation of endometrial cancer cells[17]. Still, it remains unclear why the stromal fibroblasts decreased but showed stronger effect and what mechanisms could cause normal stromal fibroblasts to transform to CAFs during endometrial tumorigenesis.

Analysis of the dynamic immune microenvironment changes during the tumorigenic process of EEC showed tendencies of decreased proportion of cytotoxic and naive CD8 lymphocyte population and increased proportion of CD4 Treg population, indicating immune escape during endometrial tumorigenesis. However, more experimental validation is needed. For example, use immunostaining or flow cytometry to quantify the composition changes of immune cells in different stages of endometrial pathology, or establish co-culture system of immune cells and organoids to functionally verify their cross-talk effect. These distinctive features of the immune microenvironment during endometrial tumorigenesis may open up other possibilities for targeted prevention or treatment strategies.

It should be mentioned here that we are unlikely to achieve access to materials from the same patient to study the evolution of normal endometrium to AEH to EEC. Alternatively, we employ individual samples from different patients. However, the sample size of scRNA-seq in this study is still limited. Given the heterogeneity of endometrial cancer in terms of the tumor microenvironment composition and the underlying molecular alterations, the interpretation of the findings in this study cannot reflect the situation of all EEC patients. Additional scRNA-seq samples and further in vitro and in vivo validation experiments are needed. Meanwhile, the endometrial cancer samples that were analyzed in our study are not representative of all endometrial cancer molecular subtypes. Only samples (NSMP and MMRd) with relatively low levels of chromosomal instability were included, and the copy number-high, as well as *POLE*mut molecular subtype, remain to be studied.

In summary, we employ scRNA-seq to profile cell atlas from normal endometrium, AEH, and EEC, which altogether represent the step-by-step development of EEC. Our study demonstrates that EEC originates from the unciliated glandular epithelial cells, and the emergence of LCN2 + /SAA1/2+cells is a feature of endometrial tumorigenesis. Finally, we describe the stromal and immune environment changes during EEC progression. Our study elucidates the evolution and characterization of cell populations in EEC development at single-cell resolution, which would facilitate endometrial cancer research and aid the development of diagnosis strategies.

## Methods

### Ethics and clinical sample collection
This study complied with all relevant ethical regulations and was approved by the Institutional Review Board of the Obstetrics and Gynecology Hospital of Fudan University (No.2021-182). All patients were fully informed of the usage of their samples for this study and provided written informed consent.

All human endometrial samples were obtained from patients at the obstetrics and gynecology hospital of Fudan University in China. Normal endometrium was obtained from patients who were subjected to hysterectomy due to uterine fibroids or adenomyosis or early cervical cancers. Samples of AEH and EEC were obtained from patients who received hysterectomy because of the aforementioned diseases. In order to prevent the effect of age on the endometrium, endometrial samples were only obtained from women of 44 to 55 years old. Because there might be difference in the cell composition in endometrium located at different part of uterine cavity, normal endometrium or endometrial lesion were collected only at the fundus of uterine cavity. All selected patients with endometrial diseases were diagnosed as indicated disease by endometrial sampling before surgery. There must be visible lesions at fundus of uterine cavity when samples for this study were taken. All specimens for single-cell sequencing were pathologically confirmed as normal endometrium or indicated endometrial diseases by professional pathologists at Obstetrics and Gynecology Hospital, based on pathologies of fast frozen sections and Hematoxylin and Eosin (H&E) slices after hysterectomy. The AEH lesions collected had no concurrent endometrial cancer. Totally, five normal endometrial tissues, five AEH tissues, and five EEC grade one tissues were enrolled in single-cell sequencing.

For co-culture assay, normal stromal fibroblasts (NF) were collected from normal endometrium, and cancer-associated fibroblasts (CAF) were collected from EEC. For immunofluorescence staining, all of the experiments were carried out on three subjects, and the paraffin-embedded sections were collected from the tissue bank in Obstetrics and Gynecology Hospital of Fudan University.

### Human endometrial tissue dissociation and preparation for scRNA sequencing
The fresh endometrial tissues were mechanically cut by scalpels as small as possible, followed by dissociating in 20 ml of endometrial digestion medium (2.5 U/ml Dispase II (Sigma), 2 mg/ml collagenase V (Sigma) and 10% FBS) with gentle shaking at 37 °C for 1–2 h. Pipetted up and down every 20 min. Till most cells in single cell status, suspended the digestion and filtered the digested cells with 40 μm cell sieves and obtained single cell suspension. The single-cell suspension was centrifuged at 300 x *g* for 3 min and then lysed red blood cells. For scRNA-seq, the percentage of the live single cells (counted by Typan blue) was over 80% in total endometrial cells before manufacturing in

the single-cell-A-Chip (10X Genomics). Then, the chip with endometrial cells was loaded in a 10X Chromium single-cell instrument (10X Genomics). The following barcoding, cDNA synthesis and library construction were followed by standard manufacturer's instructions. The qualified libraries were applied to the Illumina NovaSeq6000 platform for PE150 sequencing.

## EEC molecular classification

Molecular classification was applied according to the diagnostic algorithm for the integrated histo-molecular endometrial carcinoma classification (WHO classification of tumors, 5th edition, female genital tumours)[57]. Briefly, the coding region and the exon/intron junctions of about 20 bp were tested in *POLE. POLE* exonuclease domain mutations (EDMs) were evaluated according to the 11 mutations reported by literature[58]. Immunohistochemistry (IHC) analysis was used for evaluating MMR (mismatch repair) and p53 status, including MLH1 (DAKO-ES05), PMS2 (DAKO-EPS1), MSH2 (DAKO-FE11), MSH6 (DAKO-EP49), and p53 (DAKO-DO-7). Once there is doubt about the IHC result of MMR status, a Promega MSI Analysis System (Version 1.2) was used on Biosystems 3500 and 3500xL Genetic Analyzers (Thermo Fisher Scientific) to verify the diagnosis.

## scRNA-seq data preprocessing

Raw reads in the.fastq files of human endometrial cells were processed in the Cell Ranger Software Suite (10x Genomics Cell Ranger 4.0.0)[59] using refdata-gex-GRCh38-2020-A as reference to map reads on the human genome (GRCh38/hg38), and generated the unique molecular identifier (UMI) matrices. The Cell Ranger outputs were imported into Seurat[18] by the 'Read10X' function. Among each sample, cells with UMI counts above upper 10% are removed. Then cells with fewer than 500 UMI counts detected or >40% mitochondrial UMI counts were filtered out. Finally, genes expressed in less than 10 cells were also removed. After the quality filtering, 99,215 cells from 15 subjects were selected for the following analysis.

## Unsupervised clustering analysis

The preprocessed data were normalized and scaled using Seurat function NormalizeData, and FindVariableGenes was used to identify highly variable genes. The principal components (PCs) were estimated by RunPCA. UMAP and t-SNE dimensionality reduction were then performed by the RunUMAP and RunTSNE to place cells with similar local neighborhoods in dimension 1 to dimension 40 and visualize the distribution of cells. The cell types were annotated based on the expression pattern of differentially expressed genes (DEGs) and the well-known cellular markers from the literatures (19, 34–39). After scaled, we use FindNeighbors with dimensions 1 to dimension 40 to allocated cells. FindClusters was conducted with the parameter 'res' to adjust the resolution.

## Data integration

Seurat function FindIntegrationAnchors was used to remove batch effect from different batches and integrate single cell data (normalization.method = "LogNormalize", dims = 1:40, reduction = "cca").

## Marker identification

In order to identify signature genes in each cell type, Seurat function FindMarkers and FindAllMarkers were carried out (min.pct = 0.1, test.use = 'roc', return.thresh = 0.25).

## Spearman correlation analysis

The expression data was subsetted into different cell types and stages, and the mean expression of cell type markers were calculated to represent the expression level of its corresponding cell type. Finally, the Spearman correlation test was carried out between Normal, AEH and EEC.

## Estimation of CNVs

The R package InferCNV was used to identify somatic large-scale chromosomal copy number variation of each gene (cutoff = 0.1), using the control group as reference[60,61]. Heatmaps were then made to show copy number variation with specific cell type and segments of chromosome.

In order to inspect the copy number alteration from different cell types, separately, we adjusted the HMM prediction matrix to zero-centered, and calculated the sum of square to get cell-based level of copy number variation.

## Whole-exome sequencing (WES) for CNVs analysis

Genomic DNA was extracted and fragmented from two normal endometrial tissue samples using DNeasy Tissue Kit (Qiagen, Cat: 69504) and DNA Library Prep Kit (BioVision, Cat: K1475-12) according to the manufacturer's recommendations. Whole-exome capture was performed using an Agilent SureSelect Human All Exon 50 Mb Kit (Agilent Technologies, Santa, Clara, California, USA). WES library was sequenced as 150 bps paired-ends read by Illumina Nova-seq 6000 platform (Illumina, San Diego, California, USA). The WES sequencing data was cleaned to pass quality controls, and mapped to human reference genome (hg19/GRCh37) by Bowtie2 (V2.4.2) with no more than 2 mismatches. The mapped BAM file was sorted by SAMtools (V.1.11) and removed duplicates by Picard (V2.20.1). Control-FREEC (control-FREE Copy number caller) (V11.6) was used to identify Copy Number Variations (CNVs) based on default parameters.

## Calculation of RNA velocity

We performed the RNA velocity analysis using the Python package scVelo to inform the transition among cell types. The BAM file from Cell Ranger were used to generate count matrices of pre-mature (unspliced) and mature (spliced) abundances, using the 'velocyto' package[62], and the latter were used as the input for scVelo afterwards. The dynamical model in scVelo[63] was used, and the results were projected back to UMAP generated by Seurat.

## Pseudobulk RNA-seq based PCA and gene enrichment analysis

For PCA, we performed pseudobulk RNA-seq analysis: The UMI counts from cells of each sample were summed up, generating a pseudobulk RNA-seq expression matrix. The matrix was log2 transformed and exported for PCA analysis using plotPCA from DESeq2 package afterwards.

GO and KEGG enrichment analysis were carried out using enrichGO and enrichKEGG using the R package clusterProfiler.

## Survival analysis

Statistical analyses were performed using the R statistical environment (v4.0.4). The TPM expression data were downloaded from Uterine Corpus Endometrial Carcinoma (TCGA-UCEC) database using the R package TCGAbiolinks[64]. We included 411 samples with endometrioid endometrial cancer and 35 normal samples out of the 589 samples for downstream expression level comparison. The clinical data were obtained from UCSC Xena database. The FPKM data were converted to TPM and merged with clinical data, after which patient-barcodes with no survival data or those with meaningless survival data were removed. A total of 393 EEC patients were included for survival analysis, and the time-span of follow-up was restricted to 100 months.

The genes highly expressed in single cell data (EEC stage compared with normal stage) were verified afterward. For example, LCN2, SAA1 and SAA2 were selected for survival analysis.

The assumptions of the Cox proportional hazards model were tested using the 'cox.zph' function in the R package Survival (v3.2-13) with 0.1 as cutoff. A log-rank test was used when the cox.zph test failed.

Risk score was calculated by formula below:

$$\text{Risk Score} = \text{ExpRNA1} \ast \beta\text{RNA1} + \text{ExpRNA2} \ast \beta\text{RNA2} + \cdots + \text{ExpRNAn} \ast \beta\text{RNAn}$$

("Exp" denotes the expression level of RNAs, and "β" is the regression coefficient obtained from the multivariate Cox regression model). The median was set as the cut-off value to stratify EEC patients into low-risk and high-risk groups. 100 months was chosen as endpoint.

### Hematoxylin-eosin (H&E) staining and immunofluorescence

For histological analysis, endometrial tissues were inflated and fixed in 4% Paraformaldehyde Fix Solution at 4 °C for 24 h. After the fixation, tissues were subsequently dehydrated and embedded in paraffin. Each section was obtained at 5 μm, and was subjected to a series processes of hydration for the H&E staining and histopathological analyses. For immunofluorescence analysis, the sections were deparaffinized and rehydrated, and the antigen retrieval was accomplished in citrate buffer (10 mM, pH 6.0). After blocking for 1 h at room temperature with 5% bovine serum albumin, the endometrial sections were incubated with the primary antibodies overnight at 4°C. After removing the primary antibodies, all sections were incubated with the fluorescent secondary antibodies at room temperature for 1 h. The following primary and secondary antibodies were used: anti-LCN2 (proteintech, 1:100), anti-SAA1/2 (Abmart, 1:100), anti-DKK4 (Biorbyt, 1:100), anti-EPCAM (proteintech, clone number 2A2D5, 1:100), anti-Vimentin (abcam, clone number EPR3776, 1:200), goat anti-mouse Alexa Fluor 647 (abcam, 1:200), goat anti-rabbit Cy3 (abbkine,1:200). Images were acquired with a thunder imager DM6B (Leica) and the confocal microscopy (OLYMPUS).

### Co-culture of the stromal fibroblast cell with EEC organoids

Normal and EEC endometrial tissues were collected from the clinically resected uterus and cut as small as possible. After digesting with collagenase at 37°C for 50–60 min with gentle shaking, let the supernatant stand at room temperature for 3 minutes. Since stromal cells are disaggregated easier than epithelial cells, they would disperse into the supernatant while epithelial cells settle to the bottom. The supernatant was filtered through one or two sieves (100 μm), and the passing fluid (contain stromal cells) was collected for further experiments. The epithelial components were washed twice and collected by centrifugation, and then mixed with ice-cold Matrigel for organoid establishment. The EEC organoids used in this study are from cell lines preserved in the laboratory.

For EEC organoids culturing, the cocktail medium is consist of Advanced DMEM/F-12 supplemented with GlutaMax (Invitrogen), B plus supplement (bioGenous), 1.25 mM N-acetyl-L-cysteine, 5 mM Nicotinamide, 250 ng/ml R-spondin1 (OrganRegen), 100 ng/ml Noggin (OrganRegen), 5 ng/ml EGF (Invitrogen), 20 ng/ml FGF 10 (OrganRegen), 500 nM A83-01 and 500 nM SB202190. The medium was refreshed every 3 days.

Transwell plates (0.4 μm pore size membrane) were used for co-culture. EEC organoids were seeded in the lower well and fibroblast cells (obtained from normal endometrium of patient with uterine fibroids or endometrial cancer tissues of patient with EEC) were seeded in the upper well for 1 week. The ratio of stromal ($1 \times 10^4$ cells) to organoids ($1 \times 10^4$ cells) was 1: 1. After 1 week of co-culture, EEC organoids were collected for subsequent experiments.

### Quantification of mRNA expression levels by real-time PCR

Total RNA was extracted from cells using the RNAprep pure Micro Kit (TIANGEN), and 5× FastKing-RT SuperMix (TIANGEN) were used to reverse-transcribe RNA into cDNA. Quantitative PCR (qPCR) was performed using 2x SYBR Green qPCR master mix (Bimake), and the expression levels were normalized to GAPDH and calculated using the $2 - \Delta\Delta Ct$ method. The primers used in the study are listed as follow:

GAPDH- F: AACGGATTTGGTCGTATTG, GAPDH- R: GGAAGATGGTG ATGGGATT; ALDH1- F: GACAATGCTGTTGAATTTGCAC, ALDH1- R: AAGGATATACTTCTTAGCCCGC; SOX9- F: AGCGAACGCACATCAAGA C, SOX9- R: CTGTAGGCGATCTGTTGGGG; CD133- F: GGCAACAGC GATCAAGGAG, CD133- R: GATGGATGCACCAAGCACAG; MYC- F: CGACGAGACCTTCATCAAAAAC, MYC- R: CTTCTCTGAGACGAGCT TGG; AXIN2- F: GTCTCTACCTCATTTCCCGAGAAC, AXIN2- R: CGAGAT CAGCTCAGCTGCAA.

### Statistics and reproducibility

The statistical tests for each figure are indicated in the figure legend. Data are presented as means, standard deviation (s.d.), and standard error of mean (s.e.m.), as noted in the figure legends. All samples represent biological replicates, and the number is mentioned in the corresponding figure legend. Experiments were repeated independently with similar results to demonstrate reproducibility. Statistical analyses were performed using Graphpad Prism (V9.2). No statistical method was used to predetermine the sample size. No data were excluded from the analyses. The experiments were not randomized. The Investigators were not blinded to allocation during experiments and outcome assessment.

### Reporting summary

Further information on research design is available in the Nature Research Reporting Summary linked to this article.

### Data availability

All of the data for this manuscript have been made publicly available. The single-cell RNA sequencing data reported in this paper have been deposited in NCBI SRA database under the accession number SRP349751. WES sequencing data of two normal subjects reported in this paper have been deposited in NCBI SRA database under the accession number of SRP396178. The raw sequencing data are accessible for non-commercial purposes. TCGA-UCEC datasets (https://portal.gdc.cancer.gov/, details are described in Source data Figure 5) were also used in this study. Source data are provided with this paper.

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

## Acknowledgements

This work was supported by grants from the National Key Research and Development Program of China (2019YFC1005200 and 2019YFC1005201, X.C.) and the National Natural Science Foundation of China (82071611, X.C., and 32022022, B.Z.).

## Author contributions

X.R., B.Z., and X.C. conceived the study; X.R., Y.X., Y.W., and M.Q. performed the experiments; J.L., Y.Z., and N.J. analyzed the data; B.Z. and X.C. supervised the work; and X.R., J.L., B.Z., and X.C. wrote the manuscript.

## Competing interests

The authors declare no competing interests.

## Additional information

**Correspondence and requests** for materials should be addressed to Bing Zhao or Xiaojun Chen.

