## [Peer review file · Nature Communications]

REVIEWER COMMENTS

Reviewer #1 (Remarks to the Author): Expert in endometrial cancer genomics

In this manuscript, Ren and colleagues performed single-cell RNA-sequencing (scRNA-seq) of normal endometrium, atypical endometrial hyperplasia (AEH) and endometrioid endometrial carcinoma (EEC), and concluded that the study presented “elucidated the evolution of cell populations in endometrial cancer at single-cell resolution, which would facilitate endometrial cancer research, diagnosis, and therapeutics”.

The analysis of the progression of normal to precancerous to cancerous endometrial tissue is undoubtedly meritorious, given that this evolution is not well-understood, however there are a few shortfalls.

- Given the heterogeneity of endometrial cancer in terms of the levels/ composition of the tumor microenvironment and the underlying molecular alterations, the evolution of normal endometrium to AEH to EEC is best studied using material obtained from the same patient. The current analysis is confounded by having obtained the different tissue samples from different patients with distinct “complications” (see Table S1) and potentially varying cellular compositions.
- Only 3 patients are used for each condition (i.e. normal, AEH, EEC), which limits potential generalizability of the data to the broader endometrial cancer population, especially given the known heterogeneity of this disease.
- Was the scRNA-Seq analysis performed on fresh or frozen tissue? This is not clear from the methods. Also, was the diagnosis of atypical endometrial hyperplasia (AEH) confirmed by pathology review of the surgical specimen? Did any of the AEH cases have concurrent endometrial cancer?
- The distinct cell types were identified based on previously reported markers, which included CDN and COL6A3 for stromal fibroblasts. However, to identify stromal fibroblasts by immunofluorescence, Vimentin was used as a marker. Does the expression of the previously reported markers identify the expected cell types in the different normal and AEH tissues when using an orthogonal method such as immunofluorescence?
- On page 5, a reference should be provided for the previously reported canonical cell markers employed.
- For the CNV analysis, the normal endometrium was used as control. Were the 3 normal endometrial tissues analyzed devoid of any CNVs? Also, it is not entirely clear what the sentence “Nevertheless, the origin of these cancer cells in the normal endometrium remains unclear.” relates to (page 7, last sentence first paragraph).

- Despite the use of scRNAseq, the biological insights are somewhat limited. For example, it is not surprising that endometrial cancers are derived from epithelial rather than stromal cells.
- Overall, aside from the clear differences between the epithelial cells from the different populations (normal, AEH, and EEC), the differences in other cell subsets are rather modest. For example, relative abundance of distinct fibroblast subtypes (Figure 6E) or different immune cell subtypes (Figure 7B, E, H) is quite similar, making it difficult to draw conclusions on how the microenvironment evolves in response to tumor evolution.
- The data are presented as composite of 3 samples/condition; however, these can be influenced by the relative contribution from the cells from each patient. For example, according to Table S1, 9749 cells were analyzed from EEC subject 1, while 3645 cells were analyzed from EEC subject 3. Any of the composite analyses presented are thus potentially biased by unequal contributions from each patient.
- It is not entirely clear how the work presented would facilitate endometrial cancer “diagnosis and therapeutics”. Please elaborate.

Reviewer #2 (Remarks to the Author): Expert in single-cell RNA-seq, cancer genomics, and tumour immune microenvironment

Ren et al present a single-cell analysis of 9 women with either healthy endometrium, a pre-cancerous lesion (AEH) and endometrial cancer (EC). They attempt to address the origin of EC which remains incompletely understood. To address this, they use a series of standard analytical tools using default settings. Not surprisingly, they find an increased epithelial fraction in EC compared to healthy and AEH tissue. Surprisingly, they find that epithelium in both, AEH and EC harbor copy-number variations. They propose the EC cells derived from unciliated epithelium and find a few markers associated with these; in turn they associate this signature with cells in AEH, which may indicate that these cells in fact give rise to EC. They validate a couple of markers associated with EC using IF. Furthermore, they find that LCN2, SAA1 and SAA2 are more strongly expressed in EC in TCGA. Among fibroblasts and immune cells, there are no clear differences; they perform co-culture experiments using healthy fibroblasts or CAFs, which induce expression of various markers among cancer cells that may indicate stem-ness. Lastly, among T cells, there are very modest differences; among others, they find decreased portion of FGFBP2 + CD8 cells and argue that this might indicate impaired immune surveillance in cancer.

Overall, the rationale behind analyses presented here follows a logical order towards addressing a biological question, the manuscript is well written. However, there are several concerns with respect to the data analyses, over-interpretation of results, and inconsistencies between presented data and descriptions of that data, and minimal and incomplete experimental validation, all of which need to be addressed.

Major:

Sample size: the small sample size is a serious limitation of this study that significantly hampers the author's ability to draw robust biological conclusions. I strongly recommend that they improve the sample size, at least in the AEH and EC groups. This limitation is not intrinsic to this study, but a problem in many single-cell studies and it is important to note that increasing the cells analyzed per patient is not a replacement for improving power by increasing sample size. The expectation is not that the authors include dozens of patients per group, but rather, sufficient samples (perhaps 6 per group) to improve the borderline abilities to conduct any statistical comparisons. This is particularly problematic as the authors use the CCA implementation of Seurat to integrate data. This results in significant removal of biological variability. The authors should present an un-integrated UMAP of the data.

The presentation and description of the inferred CNV are inconsistent. The truth of the matter is that the inferred CNVs presented are extremely noisy. This raises concerns around data quality and rigor. The authors describe that Chr 1 and 10 are recurrently amplified, when really the data in Fig S2G show that one sample has 1q amp while the first sample shows deletion in 16q, and Chr 10amp is essentially not visible. In contrast, EC fibroblasts show extremely noisy CNVs, which do not appear real, and are a common artifact of the inferCNV algorithm. Furthermore, in Sample 3, there is clearly variability among quality of cancer cells on the top and the bottom of the presented data. This needs to be reconciled by for example, downsampling to the same cell quality. It is also curious that lymphocytes harbor CNVs (shown in Fig 2g) - this again raises concerns about rigor of analyses and needs to be rectified. Lastly, the authors do not present the inferCNV analysis of AEH. Can the authors do several analysis: first, to confirm the actual CNVs, they should repeat inferCNV analysis using only T cells as reference. Second, show the quality of cells for each cell examined in inferCNV. Third, show the same analysis in AEH. Fourth, perform either whole-exome or whole-genome sequencing on the corresponding bulk tissue. Lastly, the authors should cite Tirosh et al, Science, 2016, in the methods section for inferCNV (currently no ref).

The authors perform velocity analysis and conclude that unciliated cells are the source of cancer and already visible in AEH. The truth of the matter is that normal cells have a lower fraction of unciliated cells, as the authors show in Fig 1, which could fully explain this observation simply due to smaller cell numbers in this cluster. Furthermore, the proportions do not add up: if ciliated cells give rise to cancer would we not expect there to be significantly more of these in the EC compared to AEH? This issue is further amplified in their next analysis (Fig. 3) where only a tiny portion of epithelium is in fact cancerous. Are the authors suggesting that even in a fully established EC only a tiny fraction are cancer cells? This section is extremely confusion and illogical. Please clarify and rectify the inconsistencies in description and cell proportions.

How do the authors explain the fact that markers they view as tumor specific are equally or even more strongly expressed in AEH (e.g. in Fig 4)? SLPI, SAA1, SAA2, LCN2, LTF and even DKK4 all show higher

expression in AEH compared to EC. Are the IF from the corresponding tissue samples or independent tissue samples?

While the in vitro experiments using fibroblast/CAF co-cultures appear to show a difference in the expression of several genes associated with stem-ness, there is no further validation of this result by for example defining the factors that promote this gene expression or perturbations of the selected genes and their effects. Overall, this section is very preliminary and overall does not provide strong support for the overreaching claim that "converted stromal fibroblasts establish oncogenic signal niche for EC". The truth of the matter is that it looks like there is no difference at all.

All observations made in the myeloid/macrophage section are extremely limited and given the small sample size, it is unclear what their biological significance is. This needs to be improved with a larger sample size.

The FGFBP2 fraction of T cells is extremely small among the overall immune infiltration, and within the T cell cluster. In and of itself, FGFBP2 is not sufficient to determine the cellular state of these cells. Can the author determine whether these cells 1) are cycling, 2) also express other genes associated with activated FGFBP2+ T cells (e.g. CXCR3), 3) determine the cell state of T cells using previously established T cell signatures (e.g. Azizi et al., Cell, 2018; Jerby-Arnon et al, Cell, 2018)?

Reviewer #3 (Remarks to the Author): Expert in endometrial cancer genomics, single-cell RNA-seq and organoids

Summary;

This paper reports on single cell gene expression data generated from 3 subjects of normal endometrium, 3 subject with atypical endometrial hyperplasia and 3 subjects with epithelial endometrioid endometrial cancer. They report on the cellular architecture of these samples, identifying increased epithelial cells in AEH and endometrioid endometrial cancer samples. They use this data to confirm endometrioid carcinomas are derived from epithelial cells and postulate it is derived from unciliated glandular epithelial cells. They identify genes LCN2 and SAA1/2 in combination may be markers for tumorigenesis. They confirm increased protein expression of these two genes with immunofluorescence.

They also go on to analysis gene expression in other cells of the microenvironment including stromal cells and immune cells, finding only minimal differences between pathologies and showing the endometrial stromal cells may support organoid growth through changes in gene expression.

Overall opinion;

I found the manuscript well written and interesting. The identification of the combined gene signature of LCN2/SAA1/2+ as a marker for endometrial cancer is promising and the follow up IF is good quality. I do however find that the study is limited by the amount of individual samples analysed and the clinical annotation accompanying these samples. The paper relies heavily on single cell data, with much of the other subsequent analysis not strongly contributing to the interpretation of the single cell data. The transition from unciliated glandular cells to endometrial cancer is based mainly on single cell gene expression data and bioinformatic inference, which while powerful scRNA-seq data still has limitations, particularly in regards to the identification of lowly expressed genes, such as transcription factors that may play critical roles. It would require functional follow up.

Major points;

- This analysis is based on only 3 samples of each pathology, which will not capture the variation inherent between patients, particular after oncogenesis, as evidenced in PCA plot in Figure S2A which shows one sample clustering completely separately from the other samples.
- In the TCGA dataset the samples analysed are termed uterine corpus endometrial carcinoma. Within this dataset there may be many different subtypes of endometrial cancer, whereas the original analysis was performed on endometrioid samples only. The subtypes and annotated data should be made clear and if necessary analysed as subtypes separately.
- While potentially a step wise progression occurs from atypical endometrial hyperplasia to endometrioid cancer exists, it does not always occur. It is not clear if any of these samples used in this analysis will progress to endometrioid carcinoma and thus whether they can be considered as a step towards endometrial cancer. With only three samples this may represent a confounding factor.

Specific points;

- This study assesses only endometrioid endometrial carcinoma, origins may not be similar with different subtypes and this should be reflected in title and abstract.

- There appears to be a significant difference in the BMI of these patients. BMI can lead to changes in gene expression, was this accounted for in gene expression profiles?
- What was the average read depth for each cell? This is important information to understand the quality of the dataset.
- For the TCGA analysis why was the cut off value set at the median, and how did this relate to values obtained in the single cell dataset?
- For the normal patients were other endometrial pathology assessed, such as endometriosis, adenomyosis or fibroids? These have postulated to altered gene expression and cell composition of the endometrium and could be a confounding factor. Again with only three samples, this may be an issue.
- It is not clear how many samples were used for immunofluorescence, or whether an analysis of these samples supported an increase in expression. One picture does not represent consistent up regulation.
- Annotation of cells via individual markers is common but a reasonably limited method for detecting subtle variations in cell types. With genome wide gene expression data many more markers can be used to better characterise cells using software such as SingleR. For example; ACTA2 is a known marker for activated fibroblasts as well as smooth muscle cells, although in this study it is used to denote smooth muscle. I believe this is reflected in the heat map that shows a dual gene expression signature in the smooth muscle cells that may reflect two different cell types.
- What about epithelial progenitor cells. Were any of these identified in any of the tissue?
- Line 80 (Supplementary Figure S2); It is not clear to me how high variability, particularly in the EEC content verifies the reproducibility of the data. In contrast at such low sample numbers it suggests significant variation in the endometrial carcinomas that might not be captured in this study.
- Line 100; I don't think that it is surprising an increase in cancer cells is a mark of cancer. Additionally, what is meant by global cell types? – A cancerous growth of cell will influence the proportion of these cells.

- Figure 1D: There is a big increase in cells considered as others. It is mentioned there is a slight increase in macrophage, but not much other detail. It would be interesting to know what these other cells are.
- Figure 2b: In the text n mentions there was no detection of OVOL1 and OVOL2 stromal cells, however in the figure there does appear to be some cells that are positive for both
- Line 159: This is quite speculative. Did any of these patients go on to develop endometrial cancer?
- Line 176 – this argues against EH having endometrial cancer clusters.
- There is very minimal details on how the co-cultures were established. Where did the stromal cells come from. What were the patient characteristics, what was the ratio of stromal to organoids loaded/ how were the cells collected?
- Were the co-cultures performed in two replicates? This seems a small number to get sufficient data for statistical analysis?
- Line 261 – this is very minimal evidence that there is a signal niche. There is no functional validation. Very little details about cells, or how they were prepared. What were the clinical details of where the stromal cell came from?
- Using only one gene for normalisation of qPCR is no longer considered to be sufficient to account for variability. Analysis of stability should be perform prior to settling on a gene for normalisation.
- The depth of analysis on the microenvironment seems only minimal. I think the variation in stromal cells between pathologies may not have been identified because resolution may not have been sufficient. An analysis program like clustree may be useful.
- At any point was a correction performed for multiple testing. In particular figure 7B would require man instances of testing.
- Line 292: This is a big assumption from the expression of a few genes.

Figure Changes

Revised Figure	Previous Figure	Modification
Figure 1A	Figure 1A	Updated the number of cells and genes
Figure 1B	Figure 1B	Updated data
Figure 1C	Figure 1C	Updated data
Figure 1D	Figure 1D	Updated data
Figure 1F	Figure 1F	Updated data
Figure 1G	Figure 1G	Updated data
Figure 2A	Figure 2A	Updated data
Figure 2B	Figure 2B	Updated data; add percentage of labeled cells
Figure 3A	Figure 3A	Updated data
Figure 3B	Figure 3B	Updated data; rename previous 'EEC_specific' to 'oncogenic subpopulation'
Figure 3C	Figure 3C	Updated data
Figure 3D	Figure 3D	Updated data; Add ciliated cells
Figure 3E	Figure 3E	Updated data; Add result of normal samples
Figure 4A	Figure 4A	Updated data
Figure 5A	Figure 5A	Re-analyzed in TCGA EEC cases
Figure 6A	Figure 6A	Updated data
Figure 6B	Figure 6B	Updated data
Figure 6C	Figure 6C	Updated data
Figure 6D	Figure 6D	Updated data
Figure 6E	Figure 6E	Updated data
Figure 6F		New data
Figure 6G	Figure 6F	
Figure 7A	Figure 7A	Updated data
Figure 7B	Figure 7B	Updated data
Figure 7C	Figure 7C	Updated data
Figure 7D	Figure 7D	Updated data
Figure 7E	Figure 7E	Updated data
Figure 7F	Figure 7F	Updated data
Figure 7G	Figure 7G	Updated data
Figure 7H	Figure 7I	Updated data
Figure 7I	Figure 7H	Updated data
Figure S1A	Figure S1A	Updated data
Figure S2A	Figure S2A	Updated data
Figure S2B	Figure S2B	Updated data
Figure S2C	Figure S2C	Updated data
Figure S2D	Figure S2D	Updated data

Figure S2E	Figure S2E	Updated data
Figure S2F	Figure S2F	Updated data
Figure S3A	Figure S2G	Updated data
Figure S3B		New data
Figure S4	Figure S3	Updated data
Figure S5A	Figure S4A	Updated data
Figure S5B	Figure S4B	Updated data
Figure S5C	Figure S4C	Updated data
Figure S5D	Figure S4D	Updated data
Figure S5E	Figure S4E	Updated data
Figure S5F	Figure S4F	Updated data
Figure S6A		New data
Figure S6B		New data
Figure S7A	Figure S5A	Updated data
Figure S7B	Figure S5B	Updated data
Figure S7C	Figure S5C	Updated data
Figure S7D	Figure S5D	Updated data
Figure S7E	Figure S5E	Updated data
Figure S7F	Figure S5F	Updated data
Figure S8		New data
Table S1	Table S1	Updated data
Table S2	Table S2	Updated data

Reviewer #1 (Remarks to the Author): Expert in endometrial cancer genomics

In this manuscript, Ren and colleagues performed single-cell RNA-sequencing (scRNA-seq) of normal endometrium, atypical endometrial hyperplasia (AEH) and endometrioid endometrial carcinoma (EEC), and concluded that the study presented “elucidated the evolution of cell populations in endometrial cancer at single-cell resolution, which would facilitate endometrial cancer research, diagnosis, and therapeutics”.

The analysis of the progression of normal to precancerous to cancerous endometrial tissue is undoubtedly meritorious, given that this evolution is not well-understood, however there are a few shortfalls.

- Given the heterogeneity of endometrial cancer in terms of the levels/ composition of the tumor microenvironment and the underlying molecular alterations, the evolution of normal endometrium to AEH to EEC is best studied using material obtained from the same patient. The current analysis is confounded by having obtained the different tissue samples from different patients with distinct “complications” (see Table S1) and potentially varying cellular compositions.

Response:

We thank the reviewer for the constructive comments. We agree that the pathological evolution from normal endometrium to AEH then to EEC is best studied using samples obtained from the same patient. However, given that (1) freshly prepared sample is required to achieve high quality single-cell sequencing and (2) it is difficult to accurately determine the different pathological states of fresh samples, we should admit that it is difficult to precisely obtain fresh materials of normal endometrium, AEH lesion, and EEC lesion from the same patient. Moreover, concerning the priority of clinical diagnosis, multiple points and abundant materials from the same patient specimen are restricted. Therefore, we assess how normal endometrium evolves to AEH and to EEC by studying samples from different patients alternatively. Notably, to minimize the effects of individual differences and their complications on results, we increase the sample size (five cases for each condition, **New Table S1**) for analysis and reach consistent conclusions (**New Figure 1, Figure 2 and Figure 3**).

- Only 3 patients are used for each condition (i.e. normal, AEH, EEC), which limits potential generalizability of the data to the broader endometrial cancer population, especially given the known heterogeneity of this disease.

Response:

To address the reviewer’s concern, we increase the sample size from 9 patients (3 samples for each condition) to 15 patients (5 samples for each condition). Although the heterogeneity and consequent possible biases might exist, the major conclusions are consistent: EEC originates from endometrial epithelial cells but not stromal cells, and unciliated glandular epithelium is the source of EEC (**New Figure 3D**). Our finding indicates that single-cell RNA-sequencing (scRNA-seq) should be a reliable tool for investigating the possible mechanism of endometrial tumorigenesis.

- Was the scRNA-Seq analysis performed on fresh or frozen tissue? This is not clear from the methods. Also, was the diagnosis of atypical endometrial hyperplasia (AEH) confirmed by pathology review of the surgical specimen? Did any of the AEH cases have concurrent endometrial cancer?

Response:

All the scRNA-Seq analyses in this study were performed with fresh tissue. This study was conducted in a specialized Ob & Gyn hospital, and the diagnosis of each specimen was confirmed by professional pathologists. Pathology of fast-frozen resected specimen was examined by experienced pathologists to confirm its AEH diagnosis. Hematoxylin and Eosin (H&E) staining of the specimen was examined by at least two experienced pathologists for further confirmation (**New Figure S1**). Only the specimens with consistent pathological diagnoses of both frozen and H&E staining sections were included in this study. The AEH lesions collected had no concurrent endometrial cancer. We add the information in revised Methods for clarity (**Line 247-248**).

- The distinct cell types were identified based on previously reported markers, which included DCN and COL6A3 for stromal fibroblasts. However, to identify stromal fibroblasts by immunofluorescence, Vimentin was used as a marker. Does the expression of the previously reported markers identify the expected cell types in the different normal and AEH tissues when using an orthogonal method such as immunofluorescence?

Response:

DCN, COL6A3, and Vimentin have been reported to be highly expressed in the stroma of endometrium, which is in consistence with our scRNA-seq results (**New Figure 1C**). Regarding that Vimentin is a well-recognized signature marker for stromal cells, we employed Vimentin to identify stromal cells by immunofluorescence (**Figure 1E**). And the expression pattern of Vimentin shown in immunofluorescence accords with that in the scRNA-seq data.

- On page 5, a reference should be provided for the previously reported canonical cell markers employed.

Response:

The reference [1] is added (**Line 83-85**).

- For the CNV analysis, the normal endometrium was used as control. Were the 3 normal endometrial tissues analyzed devoid of any CNVs? Also, it is not entirely clear what the sentence “Nevertheless, the origin of these cancer cells in the normal endometrium remains unclear.” relates to (page 7, last sentence first paragraph).

Response:

To evaluate the CNVs of normal endometrium, we perform whole exome sequencing (WES) on two independent cases. As shown in the **New Figure S3A**, the patterns of copy number profile are stabilized, implying low CNVs.

To avoid misunderstanding, we revise the text “*Nevertheless, the origin of these cancer cells in*

endometrium remains unclear” to “Nevertheless, the origin of these cells with high CNVs in endometrial cancer remains unclear” (Line 122-123).

- Despite the use of scRNAseq, the biological insights are somewhat limited. For example, it is not surprising that endometrial cancers are derived from epithelial rather than stromal cells.

Response:

Indeed, the main goal of this work is to identify the origin of EEC, which has been a subject of debate in literatures supporting both epithelial and stromal lineages could be the origin of EEC [2]. Our data indicated that EEC originates from the endometrial epithelial cells instead of stromal cells, and the unciliated glandular epithelium is the source of EEC. Additionally, we found that LCN2+/SAA1/2+ cells were featured subpopulations of endometrial tumorigenesis. Our study therefore provides new critical evidence for the better understanding of endometrial cancer development.

- Overall, aside from the clear differences between the epithelial cells from the different populations (normal, AEH, and EEC), the differences in other cell subsets are rather modest. For example, relative abundance of distinct fibroblast subtypes (Figure 6E) or different immune cell subtypes (Figure 7B, E, H) is quite similar, making it difficult to draw conclusions on how the microenvironment evolves in response to tumor evolution.

Response:

Though there is rather modest change in the proportions of subtypes in EEC stromal cells when compared to normal endometrium (**New Figure 6E**), functional experiment indicates their potential intrinsic differences. When co-cultured with cancer-associated fibroblasts (CAF), the expression levels of stemness-related genes in the EEC organoids were higher than those co-cultured with normal stromal fibroblast (NF) (**Figure 6, F and G**), suggesting potentially distinct biological functions of fibroblast in the tumor milieu. This difference could not be distinguished in single-cell atlas, possibly due to the limited depth of sequence.

Similarly, the proportions of subtypes in immune cells of AEH and EEC do not change significantly when compared to normal endometrium (**New Figure 7B**), whereas a distinct subcellular cluster of T cells indicates immune changes of tumor microenvironment (**New Figure 7, G and I**). The proportions of cytotoxic and exhausted CD8 T cell are significantly reduced in EEC, compared to normal samples (**New Figure 7I**), and the proportion of FOXP3+ CD4 Treg lymphocytes, related to immunosuppression, is significantly increased in EEC samples (**New Figure 7I**). Moreover, we delineate subcellular cluster of macrophages in normal and diseased endometrium, dividing them into FCN1+ and SPP1+ subtypes (**New Figure 7C**). Although there is no significant change in the proportions of these subtypes in macrophages between normal, AEH, and EEC samples (**New Figure 7E**), KEGG analysis of these two distinct macrophage subsets shows that FCN1+ macrophages are related to positive regulation of cytokine production and cellular response, while SPP1+ macrophages are related to positive regulation of lymphocyte activation (**New Figure 7F**).

- The data are presented as composite of 3 samples/condition; however, these can be influenced

by the relative contribution from the cells from each patient. For example, according to Table S1, 9749 cells were analyzed from EEC subject 1, while 3645 cells were analyzed from EEC subject 3. Any of the composite analyses presented are thus potentially biased by unequal contributions from each patient.

Response:

The sample size is increased to five cases for each condition. All data from the study have been updated, and the results and major conclusions are well replicated. Although the number of cells varies among individuals, the minimum number of cells is 3097 per sample (**New Table S2**), which is within the requirements to capture cell subsets by 10x Genomics single-cell sequencing platform. In addition, the cell type components are consistent among samples at different stages of endometrial pathology (**New Figure S2F**).

- It is not entirely clear how the work presented would facilitate endometrial cancer “diagnosis and therapeutics”. Please elaborate.

Response:

Indeed, current results are insufficient to substantially support clinical treatment and diagnosis. Nevertheless, considering that AEH is the precursor lesion of EEC with high risk to develop into EEC, genes identified from oncogenic subpopulation of EEC and AEH could be characteristics of tumor progression. Further studies are needed to evaluate the role of the oncogenic subpopulation in early diagnosis of EEC. We elaborate in revised Discussion (**Line 342, 353-355, 376-378, 407-410**).

Reviewer #2 (Remarks to the Author): Expert in single-cell RNA-seq, cancer genomics, and tumour immune microenvironment

Ren et al present a single-cell analysis of 9 women with either healthy endometrium, a pre-cancerous lesion (AEH) and endometrial cancer (EC). They attempt to address the origin of EC which remains incompletely understood. To address this, they use a series of standard analytical tools using default settings. Not surprisingly, they find an increased epithelial fraction in EC compared to healthy and AEH tissue. Surprisingly, they find that epithelium in both, AEH and EC harbor copy-number variations. They propose the EC cells derived from unciliated epithelium and find a few markers associated with these; in turn they associate this signature with cells in AEH, which may indicate that these cells in fact give rise to EC. They validate a couple of markers associated with EC using IF. Furthermore, they find that LCN2, SAA1 and SAA2 are more strongly expressed in EC in TCGA. Among fibroblasts and immune cells, there are no clear differences; they perform co-culture experiments using healthy fibroblasts or CAFs, which induce expression of various markers among cancer cells that may indicate stem-ness. Lastly, among T cells, there are very modest differences; among others, they find decreased portion of FGFBP2 + CD8 cells and argue that this might indicate impaired immune surveillance in cancer.

Overall, the rationale behind analyses presented here follows a logical order towards addressing a biological question, the manuscript is well written. However, there are several concerns with respect to the data analyses, over-interpretation of results, and inconsistencies between presented data and descriptions of that data, and minimal and incomplete experimental validation, all of which need to be addressed.

Major:

Sample size: the small sample size is a serious limitation of this study that significantly hampers the author's ability to draw robust biological conclusions. I strongly recommend that they improve the sample size, at least in the AEH and EC groups. This limitation is not intrinsic to this study, but a problem in many single-cell studies and it is important to note that increasing the cells analyzed per patient is not a replacement for improving power by increasing sample size. The expectation is not that the authors include dozens of patients per group, but rather, sufficient samples (perhaps 6 per group) to improve the borderline abilities to conduct any statistical comparisons. This is particularly problematic as the authors use the CCA implementation of Seurat to integrate data. This results in significant removal of biological variability. The authors should present an un-integrated UMAP of the data.

Response:

We thank the reviewer for the constructive comments. Accordingly, we increase the sample size to 15 (5 samples for normal endometrium, EAH, and EEC each) (**New Table S1**). The results throughout the manuscript are updated, and the conclusions are well replicated.

We agree that integrating data using Seurat's CCA implementation may eliminate biological variability to some extent. We present the individual UMAP for each sample and the bubble map of the marker genes, supporting that the clustering results of the independent and integrated analyses are consistent (normal subjects in **Attached Figure 1**, AEH subjects in **Attached Figure 2** and EEC subjects in **Attached Figure 3**). Notably, the new subpopulations we found by integrated analyses

(showing in **Figure 3A**) only exist in AEH and EEC samples.

Attached Figure 1 | Un-integrated UMAP of normal endothelium samples. (A) UMAP showing main cell types of normal endothelial tissues from 5 independent samples (right panel) and the new subpopulations (C14 and C28), which we found in Figure 3A (left panel). (B) Bubble map showing the expression patterns of canonical markers of each cell type. Each dot represents a gene, of which the color saturation indicates the average expression level, and the size indicates the percentage of cells expressing the gene.

Attached Figure 2 | Un-integrated UMAP of AEH samples. (A) UMAP showing main cell types of AEH tissues from 5 independent samples (right panel) and the new subpopulations (C14 and C28), which we found in Figure 3A (left panel). **(B)** Bubble map showing the expression patterns of canonical markers of each cell type. Each dot represents a gene, of which the color saturation indicates the average expression level, and the size indicates the percentage of cells expressing the gene.

Attached Figure 3 | Un-integrated UMAP of EEC samples. (A) UMAP showing main cell types of EEC tissues from 5 independent samples (right panel) and the new subpopulations (C14 and C28), which we found in Figure 3A (left panel). **(B)** Bubble map showing the expression patterns of canonical markers of each cell type. Each dot represents a gene, of which the color saturation indicates the average expression level, and the size indicates the percentage of cells expressing the gene.

The presentation and description of the inferred CNV are inconsistent. The truth of the matter is that the inferred CNVs presented are extremely noisy. This raises concerns around data quality and rigor. The authors describe that Chr 1 and 10 are recurrently amplified, when really the data in Fig S2G show that one sample has 1q amp while the first sample shows deletion in 16q, and Chr 10amp is essentially not visible. In contrast, EC fibroblasts show extremely noisy CNVs, which do not appear real, and are a common artifact of the inferCNV algorithm. Furthermore, in Sample 3, there is clearly

variability among quality of cancer cells on the top and the bottom of the presented data. This needs to be reconciled by for example, downsampling to the same cell quality. It is also curious that lymphocytes harbor CNVs (shown in Fig 2g) - this again raises concerns about rigor of analyses and needs to be rectified. Lastly, the authors do not present the inferCNV analysis of AEH. Can the authors do several analysis: first, to confirm the actual CNVs, they should repeat inferCNV analysis using only T cells as reference. Second, show the quality of cells for each cell examined in inferCNV. Third, show the same analysis in AEH. Fourth, perform either whole-exome or whole-genome sequencing on the corresponding bulk tissue. Lastly, the authors should cite Tirosh et al, Science, 2016, in the methods section for inferCNV (currently no ref).

Response:

We appreciate the reviewer's suggestions. We tried to repeat inferCNV analysis using T cells as reference, but found quite a lot noisy CNVs in normal epithelial cells and stromal cells (**Attached Figure 4**). To evaluate the CNVs of the normal endometrium, we perform whole exome sequencing (WES) on two independent normal endometrial samples. The patterns of copy number profile are stabilized in all chromosomes (**New Figure S3A**), implying low CNVs in cells of normal endometrial. CNV spots we found in normal epithelial and stromal cells when using T cells as reference were possibly due to cell-specific expression patterns and differences in genes expressed. Therefore, we used the same cell type for reference (normal) and observation (tumor).

We reconcile the same cell quality according to the reviewer's suggestion, showing the cell quality of epithelial cells (**Attached Figure 5** below) and fibroblast (**Attached Figure 6** below) in each group. In fact, the updated results are in line with previous ones, *i.e.* epithelial cells of AEH and EEC display the highest CNVs patterns (**New Figure 1G**). Based on the updates, we address that high CNVs occur on chromosome 1, 8, or 10 in the epithelial cells of EEC samples (**New Figure S3B**), which is in consistence with the report from Levine and the Cancer Genome Atlas Research Network [3].

The same method is used to evaluate the CNVs of the epithelial and stromal cells in AEH samples. A lot CNVs are observed in the epithelial cells of AEH samples (**New Figure 1G and Figure S3B**).

We cite the literature Tirosh *et al.* in revised Methods (**Line 490**).

Attached Figure 4 | Single cell copy-number variation analysis of epithelial cells and stromal fibroblasts in normal endometrium. Representative CNV heatmaps from inferCNV were applied in epithelial cells (**A**) and stromal fibroblasts (**B**) of five independent normal subjects. T cells were defined as baseline.

Attached Figure 5 | Quality control of epithelial cells. Mitochondrial counts ratio in normal (A), AEH (C), and EEC (E) epithelial cells, and the total number of molecules detected within a cell from normal (B), AEH (D), and EEC (F) epithelial cell groups.

Attached Figure 6 | Quality control of fibroblasts. Mitochondrial counts ratio in normal (A), AEH (C), and EEC (E) stromal cells, and the total number of molecules detected within a cell from normal (B), AEH (D), and EEC (F) stromal cell groups.

The authors perform velocity analysis and conclude that unciliated cells are the source of cancer and already visible in AEH. The truth of the matter is that normal cells have a lower fraction of unciliated cells, as the authors show in Fig 1, which could fully explain this observation simply due to smaller cell numbers in this cluster. Furthermore, the proportions do not add up: if ciliated cells give rise to cancer would we not expect there to be significantly more of these in the EC compared to AEH? This issue is further amplified in their next analysis (Fig. 3) where only a tiny portion of epithelium is in fact cancerous. Are the authors suggesting that even in a fully established EC only a tiny fraction are cancer cells? This section is extremely confusion and illogical. Please clarify and rectify the inconsistencies in description and cell proportions.

Response:

Sorry for the confusion. The origin/source of cancer cells cannot be inferred simply by the evidence of increasing cell fraction, and the heterogeneity in the same cell type makes it even harder. Our study elucidates a subtype of epithelium that unciliated glandular cells are most likely to be the origin of cancer.

The new subpopulations rise specifically in unciliated epithelial subtype of AEH and EEC by unsupervised clustering, but not in ciliated epithelial subtype (**New Figure 3A**). RNA velocity shows the movement of the unciliated cells towards a ciliated fate in both AEH and EEC samples (**New Figure 2A**). The results indicate ciliated cells are unlikely to be the source of unciliated cells in endometrial lesions.

Through further analysis of the unciliated epithelium, we find that the common subpopulations appear in both EEC (**New Figure 3B**) and its precursor lesion AEH (**New Figure 3E**), representing a commonality of inherent variation among endometrial lesions. To avoid misunderstanding, we re-define previous “EEC-specific” as “oncogenic subpopulation” (**Line 178**).

In **Figure 3B**, we label glandular cells, luminal cells, and oncogenic subpopulation, while “others” were cancer epithelial cells derived from different EEC samples. Therefore, this relatively small portion of epithelium represents a subtype of cancer cells.

In addition, RNA velocity estimation shows a strong directional flow from a group of unciliated glandular cells to the oncogenic subpopulation in EEC (**New Figure 3D**). Similar trajectory of epithelium is displayed in AEH as well (**New Figure S6B**), supporting that the tumor epithelial cells most likely originate from unciliated glandular cells.

Taken together, we capture oncogenic subpopulation of unciliated epithelium as an indication of endometrial cancerous progression, and propose that the unciliated glandular epithelial cells might be the source of endometrial carcinoma.

How do the authors explain the fact that markers they view as tumor specific are equally or even more strongly expressed in AEH (e.g. in Fig 4)? SLPI, SAA1, SAA2, LCN2, LTF and even DKK4 all show higher expression in AEH compared to EC. Are the IF from the corresponding tissue samples or independent tissue samples?

Response:

We first identified the oncogenic subpopulation by comparing normal and EEC unciliated cells, and then explored the specific markers (**New Figure 3B and New Figure S5F**). To address whether the

oncogenic subpopulation appears in EEC precursor lesion, we re-cluster the unciliated epithelium groups from AEH samples and use TransferData function to capture the oncogenic subpopulation. We find that the oncogenic subpopulation presents in AEH samples with high expression of the same signature genes (**New Figure 3E and New Figure S6A**). Further immunofluorescence of LCN2 and SAA1/2 confirms that these signature genes indeed express in AEH samples but not in normal endometrial tissues (**New Figure 3E and Figure 4B**). In sum, these signature genes mark an emerging distinctive unciliated epithelium subpopulation (oncogenic subpopulation) in endometrial precancerous and cancer cells, which would provide valuable information in future early diagnosis of EEC. The underlying mechanism of higher expression levels in AEH than EEC needs further investigation.

Immunofluorescence of LCN2 and SAA1/2 were performed on independent tissue samples.

While the in vitro experiments using fibroblast/CAF co-cultures appear to show a difference in the expression of several genes associated with stem-ness, there is no further validation of this result by for example defining the factors that promote this gene expression or perturbations of the selected genes and their effects. Overall, this section is very preliminary and overall does not provide strong support for the overreaching claim that "converted stromal fibroblasts establish oncogenic signal niche for EC". The truth of the matter is that it looks like there is no difference at all.

Response:

We add the scheme of co-culture system in **New Figure 6F**. EEC organoids are seeded with Matrigel in the lower well and fibroblasts are seeded in the upper well, mimicking the microenvironment provided by fibroblasts. We demonstrated that stromal fibroblasts from cancer donors had stronger ability to promote upregulation of stemness-related genes in EEC organoids, than those from normal donors. Our lab is carrying on extensive works to determine the underlying mechanisms.

We modify the overreaching claim to "*Therefore, our data suggest that all subtypes of stromal fibroblasts might be indispensable in both normal endometrium and EEC.*" (**Line 269-271**).

All observations made in the myeloid/macrophage section are extremely limited and given the small sample size, it is unclear what their biological significance is. This needs to be improved with a larger sample size.

Response:

The data are updated to increased sample size (five cases for each group). Although the differences in myeloid/macrophage cell subsets are rather modest, we deconstruct subcellular clusters of macrophages and lymphocytes in AEH and EEC lesions. We annotate different subtypes of macrophage and delineate the states of lymphocytes in the immune microenvironment of normal endometrium, AEH tissue, and EEC tissue (**New Figure 7C and 7G**). Although our study provides the immune atlas in AEH and EEC, the distinctive immune microenvironment features in endometrial tumorigenesis and their pathological functions need further research.

The FGFBP2 fraction of T cells is extremely small among the overall immune infiltration, and within the T cell cluster. In and of itself, FGFBP2 is not sufficient to determine the cellular state of these cells. Can the author determine whether these cells 1) are cycling, 2) also express other genes

associated with activated FGFBP2+ T cells (e.g. CXCR3), 3) determine the cell state of T cells using previously established T cell signatures (e.g. Azizi et al., Cell, 2018; Jerby-Arnon et al, Cell, 2018)?

Response:

According to the reviewer's suggestion, we re-define the cell state of T cells according to the marker genes reported in the recommended references, and update the data in **New Figure 7**. Using the T cell signatures, we classify CD8+ T cells as cytotoxic, exhausted, and naïve, and CD4+ T cells as exhausted, naive, and regulatory (**New Figure 7G, 7H and New Figure S8**).

We also identify a subset of T cells that are positive for FGFBP2 (see **Attached Figure 7A** below). However, cells of this subset do not express another related gene CXCR3, though they highly expressed cytotoxic markers (GZMA, GZMB, GNLY, and GZMH) (see **Attached Figure 7B-F** below), we thus assign them to the cytotoxic CD8 T cell subset.

Attached Figure 7 | Expression pattern of canonical marker genes in T cells. CXCR3 is a reported gene associated with activated FGFBP2+ T cells. GZMA, GZMB, GNLY, and GZMH are cytotoxic T cell markers.

Reviewer #3 (Remarks to the Author): Expert in endometrial cancer genomics, single-cell RNA-seq and organoids

Summary;

This paper reports on single cell gene expression data generated from 3 subjects of normal endometrium, 3 subject with atypical endometrial hyperplasia and 3 subjects with epithelial endometrioid endometrial cancer. They report on the cellular architecture of these samples, identifying increased epithelial cells in AEH and endometrioid endometrial cancer samples. They use this data to confirm endometrioid carcinomas are derived from epithelial cells and postulate it is derived from unciliated glandular epithelial cells. They identify genes LCN2 and SAA1/2 in combination may be markers for tumorigenesis. They confirm increased protein expression of these two genes with immunofluorescence.

They also go on to analysis gene expression in other cells of the microenvironment including stromal cells and immune cells, finding only minimal differences between pathologies and showing the endometrial stromal cells may support organoid growth through changes in gene expression.

Overall opinion;

I found the manuscript well written and interesting. The identification of the combined gene signature of LCN2/SAA1/2+ as a marker for endometrial cancer is promising and the follow up IF is good quality. I do however find that the study is limited by the amount of individual samples analysed and the clinical annotation accompanying these samples. The paper relies heavily on single cell data, with much of the other subsequent analysis not strongly contributing to the interpretation of the single cell data. The transition from unciliated glandular cells to endometrial cancer is based mainly on single cell gene expression data and bioinformatic inference, which while powerful scRNA-seq data still has limitations, particularly in regards to the identification of lowly expressed genes, such as transcription factors that may play critical roles. It would require functional follow up.

Response:

We thank the reviewer for the constructive comments. To address the reviewer's concern, we increase the sample size from 9 patients (3 samples for each condition) to 15 patients (5 samples for each condition). The major conclusions are consistent: EEC originates from endometrial epithelial cells but not stromal cells, and unciliated glandular epithelium is the source of EEC (**New Figure 3D**). Our work provides preliminary evidence that combined gene signature of LCN2/SAA1/2+ might be a marker for endometrial cancer.

We completely agree with the reviewer that more functional validation is meaningful. Further research is underway and will be presented in our next paper.

Major points;

- This analysis is based on only 3 samples of each pathology, which will not capture the variation inherent between patients, particular after oncogenesis, as evidenced in PCA plot in Figure S2A which shows one sample clustering completely separately from the other samples.

Response:

We increase the sample size to 5 samples for each condition and update the data throughout the manuscript. Despite the heterogeneity among EEC samples (**New Figure S2A**), we do capture common subpopulation in all EEC samples (**New Figure 3B and New Figure S5A**). This indicates a commonality of inherent variation among different EEC samples. Furthermore, we reveal the cellular features and cell origin of the oncogenic subpopulations, which will provide valuable information for better understanding of EEC development.

- In the TCGA dataset the samples analysed are termed uterine corpus endometrial carcinoma. Within this dataset there may be many different subtypes of endometrial cancer, whereas the original analysis was performed on endometrioid samples only. The subtypes and annotated data should be made clear and if necessary analysed as subtypes separately.

Response:

According to the reviewer's suggestion, we screen the data of EEC from TCGA database for analysis (Method section, **Line 511-513**). Based on the comparison of mRNA expression levels, *LCN2*, *SAA1*, and *SAA2* exhibit significantly upregulated expression in EEC cases compared to normal samples (**New Figure 5A**), which is consistent with our scRNA-seq results (**New Figure 4A**).

- While potentially a step wise progression occurs from atypical endometrial hyperplasia to endometrioid cancer exists, it does not always occur. It is not clear if any of these samples used in this analysis will progress to endometrioid carcinoma and thus whether they can be considered as a step towards endometrial cancer. With only three samples this may represent a confounding factor.

Response:

Indeed, we are unable to know whether these AEH will develop into endometrioid carcinoma. Given that AEH patients have a high risk of disease progression or simultaneous EC, total hysterectomy is recommended for most patients with AEH. Alternatively, to address how normal endometrium transits to AEH then to EEC, we employ individual samples from different patients. The updated data of increased sample size are consistent with previous conclusions, which indicates that our results would, to a certain extent, represent the cellular characteristics of normal endometrium, AEH, and EEC.

Specific points;

- This study assesses only endometrioid endometrial carcinoma, origins may not be similar with different subtypes and this should be reflected in title and abstract.

Response:

We appreciate the reviewer's suggestion. In this study, we did focus on the evolution and characteristics of the endometrioid endometrial carcinoma. We revise the title to "*Single-cell transcriptomic atlas highlights origin and pathological process of human endometrioid endometrial carcinoma*".

- There appears to be a significant difference in the BMI of these patients. BMI can lead to changes in gene expression, was this accounted for in gene expression profiles?

Response:

Based on individual BMI levels, we divide the patients into two groups, BMI ≥ 29 and BMI < 29 . Principal component analysis (PCA) shows no obvious correlation between BMI and expression profiles of epithelial or stromal cells (**Attached Figure 8A**).

Attached Figure 8 | Principal component analysis (PCA) shows no obvious correlation between BMI and expression profiles of epithelial or stromal cells. Red circle indicates samples with BMI ≥ 29 ; the rest are samples with BMI < 29 .

- What was the average read depth for each cell? This is important information to understand the quality of the dataset.

Response:

We add a column 'Mean reads per cell' in **Table S2** to show the average read depth of each cell in each sample.

- For the TCGA analysis why was the cut off value set at the median, and how did this relate to values obtained in the single cell dataset?

Response:

Regarding the gene expression may highly skew, quantile is frequently used as cutoff point for grouping. We tried different cutoffs at a moderate level, and the results were quite similar. So, we used the median as the cutoff value.

- For the normal patients were other endometrial pathology assessed, such as endometriosis, adenomyosis or fibroids? These have postulated to altered gene expression and cell composition of the endometrium and could be a confounding factor. Again with only three samples, this may be an issue.

Response:

The normal patients included in our study do have accompanying disease, such as adenomyosis, fibroids or cervical cancer. We cannot completely exclude the potential effects of concomitant diseases. Therefore, we increase the sample size to strengthen the results and conclusions.

- It is not clear how many samples were used for immunofluorescence, or whether an analysis of these samples supported an increase in expression. One picture does not represent consistent up regulation.

Response:

We performed immunofluorescence in three independent samples and showed the representative images in **Figure 4B**. The information is added in figure legend (**Line 811**).

- Annotation of cells via individual markers is common but a reasonably limited method for detecting subtle variations in cell types. With genome wide gene expression data many more markers can be used to better characterise cells using software such as SingleR. For example; ACTA2 is a known marker for activated fibroblasts as well as smooth muscle cells, although in this study it is used to denote smooth muscle. I believe this is reflected in the heat map that shows a dual gene expression signature in the smooth muscle cells that may reflect two different cell types.

Response:

For cell-type annotation of single-cell transcriptomic atlas, we mainly referred to the definition of endometrial cell types[1] and employed three to eight reported markers for each cell group (**New Figure 1C**). The annotations of fibroblasts and smooth muscle cells in our study are credible.

- What about epithelial progenitor cells. Were any of these identified in any of the tissue?

Response:

We do not find potential clusters of epithelial progenitors by inquiring the reported endometrial stem cell marker AXIN2. As shown in the following **Attached Figure 9**, AXIN2 positive cells are scattered in the epithelial cells. However, none of the normal epithelium, AEH epithelium and EEC epithelium forms a distinct population of AXIN2 cells. Moreover, compared to normal samples, the proportion of AXIN2 cells in epithelial cells decreases in AEH samples, while there is no significant change in EEC samples (the following **Attached Figure 9D**). Therefore, it is difficult to determine the epithelial progenitor cluster in endometrium.

Attached Figure 9 | Patterns of AXIN2 positive cells in endometrium. (A) The UMAP showed the pattern of AXIN2 positive cells in normal endometrial epithelial cells. (B) The UMAP showed the pattern of AXIN2 positive cells in AEH epithelial cells. (C) The UMAP showed the pattern of AXIN2 positive cells in EEC epithelial cells. (D) The proportion of AXIN2+ cells in epithelial cells at different stages of endometrial pathology. Data are mean \pm SEM based on five independent biological replicates. Significance was evaluated by comparing with normal group (*t*-test; no significance (n.s.), $p < 0.01$ (**)).

- Line 80 (Supplementary Figure S2); It is not clear to me how high variability, particularly in the EEC content verifies the reproducibility of the data. In contrast at such low sample numbers it suggests significant variation in the endometrial carcinomas that might not be captured in this study.

Response:

The PCA shows the consistency of biologically replicated normal endometrial tissue, supporting the repeatability of the data. We modify the sentence to “Principal component analysis (PCA) showed that the normal endometrial samples were clustered together, while the AEH and EEC samples were scattered (Supplemental Figure 2A), representing transcriptome differences in normal endometrial, AEH and EEC tissues.” (Line 75-77).

Increasing the sample size to five EEC samples, we also find the oncogenic subpopulation in each EEC samples (New Figure 3B and New Figure S5A), indicating a commonality of inherent variation among different EEC samples.

- Line 100; I don’t think that it is surprising an increase in cancer cells is a mark of cancer.

Additionally, what is meant by global cell types? – A cancerous growth of cell will influence the proportion of these cells.

Response:

Our data showed a gradual increase in the proportion of epithelial cells from normal to AEH to EEC (New Figure 1D). We change “global cell types” to “all cell types” to avoid confusion (Line 96).

- Figure 1D: There is a big increase in cells considered as others. It is mentioned there is a slight increase in macrophage, but not much other detail. It would be interesting to know what these other cells are.

Response:

In previous Figure 1D, we defined cells beyond the epithelium and stroma as other cells, including lymphocytes, macrophages, and endothelial cells. Although there was a tendency to increase in the average proportion of “Others” cell type from normal to endometrial lesion tissues, the *t*-test analysis indicated that there was no significant difference compared to normal ones.

In order to more precisely show the pattern of changes in the proportion of epithelial and stromal cells, we modify the presentation to display the proportional changes between epithelial and stroma intuitively (New Figure 1D).

The detailed analysis results of other cells (macrophages and lymphocytes) are shown in Figure 7. We annotate different subtypes of macrophage and delineated the states of lymphocytes in the immune microenvironment of normal endometrium, AEH tissue and EEC tissue (New Figure 7C and 7G).

- Figure 2b: In the text n mentions there was no detection of OVOL1 and OVOL2 stromal cells, however in the figure there does appear to be some cells that are positive for both

Response:

We analyze the number of OVOL1 and OVOL2 positive cells in stromal cells group, which accounts for only 0%-0.06% (New Figure 2B). We propose that such a tiny portion is difficult to achieve mesenchymal-epithelial transformation. Moreover, these positive cells only express low levels of OVOL1 and OVOL2, and do not form clusters, indicating that they do not have a consistent gene expression pattern.

- Line 159: This is quite speculative. Did any of these patients go on to develop endometrial cancer?

Response:

Since patients diagnosed with AEH had total hysterectomies, we are unable to know whether these AEH patients will indeed progress to endometrioid carcinoma. We modify the sentence to “*Based on these results, we speculated that new clusters (14 and 28) arising in AEH and EEC were the signal of the development of EEC, and most probably derived from normal unciliated epithelium.*” (Line 159-161).

- Line 176 – this argues against EH having endometrial cancer clusters.

Response:

We thank the reviewer for pointing this issue out. To avoid misunderstanding, we re-define the previous “EEC-specific” group to “oncogenic subpopulation” (**Line 178**). We first identified the oncogenic subpopulation by comparing normal and EEC unciliated cells, and then explored the specific markers (**New Figure 3B and New Figure S5F**). To address whether the oncogenic subpopulation appears in EEC precursor lesion, we re-cluster the unciliated epithelium groups from AEH samples and use TransferData function to capture the oncogenic subpopulation. We find that the oncogenic subpopulation presents in AEH samples with high expression of the same signature genes (**New Figure 3E and New Figure S6A**). Further immunofluorescence of LCN2 and SAA1/2 confirms that these signature genes indeed express in AEH samples but not in normal endometrial tissues (**New Figure 3E and Figure 4B**).

Taken together, our data show that the oncogenic subpopulation appears in AEH and EEC, but not in normal endometrium (**New Figure 3B and 3E**), which indicates that the oncogenic subpopulation might be associated with endometrial cancer progression.

- There is very minimal details on how the co-cultures were established. Where did the stromal cells come from. What were the patient characteristics, what was the ratio of stromal to organoids loaded/ how were the cells collected?

Response:

We add the scheme of co-culture system in **New Figure 6F**. EEC organoids are seeded with Matrigel in the lower well and fibroblasts are seeded in the upper well, mimicking the microenvironment provided by fibroblasts. Normal stromal cells are from endometrial tissue of two patients with uterine fibroids and cancer-associated fibroblasts are from endometrial tissue of two EEC patients (**New Table S1**). The ratio of stromal (1×10^4 cells) to organoids (1×10^4 cells) was 1: 1. After co-culture for one week, EEC organoids were collected for total RNA extraction experiment. We add the information in revised Methods (**Line 561**).

- Were the co-cultures performed in two replicates? This seems a small number to get sufficient data for statistical analysis?

Response:

In fact, we performed three independent experiments for statistics, with fibroblasts obtained from two individual pairs of normal and cancer patients (**New Table S1**). We correct the error in Figure Legend (**Figure 6G, Line 843-844**).

- Line 261 – this is very minimal evidence that there is a signal niche. There is no functional validation. Very little details about cells, or how they were prepared. What were the clinical details of where the stromal cell came from?

Response:

As mentioned above, we provide scheme with more experimental details for better understanding.

When co-cultured with cancer-associated stromal fibroblasts (CAF), the expression level of stemness-related genes in the EEC organoids was higher than that co-cultured with normal stromal fibroblast (NF) (**Figure 6G**), suggesting potentially distinct biological functions of fibroblast in the tumor milieu. NF cells were derived from endometrial tissue of cervical cancer patients and CAFs were derived from endometrial tissue of EEC patients. Briefly, normal and EEC endometrial tissues were collected from the clinically resected uterus and cut as small as possible. After digesting with collagenase at 37°C for 50-60 min with gentle shaking, let the supernatant stand at room temperature for 3 minutes. Since stromal cells are disaggregated easier than epithelial cells, they would disperse into the supernatant while epithelial cells settle to the bottom. The supernatant was filtered through one or two sieves (100µm), and the passing fluid (contain stromal cells) was collected for further experiments (Line 544-549). We add the information in **New Table S1** and revised Method (**Line 559-560**).

- Using only one gene for normalisation of qPCR is no longer considered to be sufficient to account for variability. Analysis of stability should be performed prior to settling on a gene for normalisation.

Response:

We tested three internal control genes (GAPDH, H3 and HPRT1) for qPCR normalization, and found that they gave comparable results in human endometrium tissues. Therefore, we kept using GAPDH for qPCR normalization throughout the study.

- The depth of analysis on the microenvironment seems only minimal. I think the variation in stromal cells between pathologies may not have been identified because resolution may not have been sufficient. An analysis program like cluster may be useful.

Response:

Indeed, the resolution is not sufficient. Due to the unsaturation of single cell sequencing data, only marker genes with high expression levels can be detected. We tried to further subdivide these subsets, but the number of cells and the depth of sequencing based on our data were limited, so we have not yet distinguished the different subsets between normal and EEC stromal fibroblasts.

- At any point was a correction performed for multiple testing. In particular figure 7B would require man instances of testing.

Response:

We did perform a correction for multiple testing. And the updated data increasing the sample size to 15 (five cases for each condition) are in consistence. (**New Figure 7B**).

- Line 292: This is a big assumption from the expression of a few genes.

Response:

We delete this assumption to avoid overreaching (previous **Line 292**).

References

1. Wang, W., et al., *Single-cell transcriptomic atlas of the human endometrium during the menstrual cycle*. Nat Med, 2020. **26**(10): p. 1644-1653.
2. Cervello, I., et al., *Stem cells in human endometrium and endometrial carcinoma*. Int J Gynecol Pathol, 2011. **30**(4): p. 317-27.
3. Cancer Genome Atlas Research, N., et al., *Integrated genomic characterization of endometrial carcinoma*. Nature, 2013. **497**(7447): p. 67-73.

REVIEWERS' COMMENTS

Reviewer #1 (Remarks to the Author):

In this revised manuscript, Ren and colleagues addressed many of the comments made, and increased the sample size from each condition 3 to 5.

- Even with the increased sample size, however, the potential generalizability of the dataset to the broader endometrial cancer population, especially given the known heterogeneity of this disease, may still be limited.
- Since the TCGA study, endometrioid endometrial cancer (EEC) is no longer perceived as a single entity. EECs are classified into four molecular subtypes (i.e., POLE, MSI, copy number-low and copy number-high) associated with distinct clinical behavior. The authors ought to specify the molecular subtype of the 5 EEC studied. For this, a well described surrogate (ProMisE) is available.
- The copy number variants differ greatly between EECs from different molecular subtypes. Is there a difference in the distribution of CNVs as shown in Figure 1G according to subtype?
- Page 5, median detected genes per cell. Please provide the range.

Reviewer #2 (Remarks to the Author):

The authors have done a great job with addressing my concerns in the revised manuscript.

In particular, they increased the sample size, updated analyses, and de-emphasized or toned down sections that may have had overreaching claims.

In the current form, I find the work to be useful for the community.

Reviewer #3 (Remarks to the Author):

The major update to the paper is the increase in sample size from 3 to 5. It is still a relatively small sample size, but I believe the manuscript is well written and care is taken not to over-interpret the results and as such believe that the authors have answered my queries. Some minor points mentioned below remain.

Major point 2: It is not clear from their response if they have stratified by subtype. The data from the TCGA analysis should not be used as a whole as it may include other subtypes that are non-endometrioid endometrial carcinoma e.g. Squamous cell carcinoma, small cell carcinoma and serous carcinoma. It should be confirmed that this has been checked and non-endometrioid cancers excluded from the analysis. If they are present they are likely to only be small numbers, but should still be excluded.

Figure 3D- I may have misinterpreted, but the authors suggest that carcinogenesis is derived from unciliated glandular cells. I cannot see the unciliated glandular cells in this figure. Are these the same as the glandular cells?

In response to the question about other pathology. It is appreciated that it is very unlikely to have true "controls" and some sort of pathology will exist, although these confounding factors should be mentioned in the discussion. Even with 5 sample numbers, these are relatively small and given these pathologies are postulated to change endometrial cell composition and gene expression in the endometrium it could influence the results and should be mentioned in the discussion.

Figure Changes

Revised Figure	Previous Figure	Modification
Figure 3d	Figure 3d	Modify the figure caption.
Figure S3a		New data.
Figure S3b	Figure S3a	
Figure S3c	Figure S3b	
Table S1	Table S1	Add molecular subtypes of EEC samples.
Table S2	Table S2	Add the gene range per cell of each sample

Reviewer #1 (Remarks to the Author):

In this revised manuscript, Ren and colleagues addressed many of the comments made, and increased the sample size from each condition 3 to 5.

- Even with the increased sample size, however, the potential generalizability of the dataset to the broader endometrial cancer population, especially given the known heterogeneity of this disease, may still be limited.

Response:

We thank the reviewer's comments. To illustrate the limitations of this study, we add a statement in the discussion section, "*However, the sample size of scRNA-seq in this study is still limited. Given the heterogeneity of endometrial cancer in terms of the tumor microenvironment composition and the underlying molecular alterations, the interpretation of the findings in this study cannot reflect the situation of all EEC patients. Additional scRNA-seq samples and further in vitro and in vivo validation experiments are needed.*" (Lines 405-409)

- Since the TCGA study, endometrioid endometrial cancer (EEC) is no longer perceived as a single entity. EECs are classified into four molecular subtypes (i.e., POLE, MSI, copy number-low and copy number-high) associated with distinct clinical behavior. The authors ought to specify the molecular subtype of the 5 EEC studied. For this, a well described surrogate (ProMisE) is available.

Response:

We thank the reviewer for the constructive comments. To specify the molecular subtype of the 5 EEC samples, we performed *POLE* sequencing and immunohistochemical staining of mismatch repair protein (MLH1, MSH2, MSH6 and PMS2) and p53 protein. Results showed that EEC-subject 1, EEC-subject 2 and EEC-subject 3 are classified into copy number-low subtype. And EEC-subject 4 and EEC-subject 5 are classified into MSI subtype. The information of molecular subtype is added in **New Table S1. (Supplementary Information)**

- The copy number variants differ greatly between EECs from different molecular subtypes. Is there a difference in the distribution of CNVs as shown in Figure 1G according to subtype?

Response:

To verify whether the distribution of CNVs varies by subtype, we perform a CNV analysis on the EEC samples case by case. According to the molecular subtypes, EEC-subject 1, 2 and 3 are classified into copy number-low subtype, whereas EEC-subject 4 and 5 are classified into MSI subtype. As shown in the following **Attached Figure 1**, there is no obvious difference in the distribution of CNVs according to subtype. The statement and results are added in the text, "*The distribution of CNVs in these five EEC samples did not show difference according to molecular subtypes (Supplemental Figure 3a).*" (Lines 113-115).

Attached Figure 1| Copy number variation score of EEC subjects case by case. According to the molecular subtypes, EEC-subject 1, 2 and 3 are classified into copy number-low subtype, whereas EEC-subject 4 and 5 are classified into MSI subtype.

• Page 5, median detected genes per cell. Please provide the range.

Response:

We add the range of detected genes per cell in each sample in **New Table S2** (Attached below).

Table S2. The single-cell RNA-seq dataset of the donors enrolled in the study.

Patient ID	Estimated Number of Cells	Cell Number After Quality Filter	Median Genes Per Cell (Range)	Median UMI Counts Per Cell	Mean reads Per Cell	Reads Mapped Confidently to Genome	Reads Mapped Confidently to Transcriptome
Normal -subject1	10,845	10,474	2,540 (307~7,736)	6,980	33,029	92.3%	55.1%
Normal -subject2	8,013	7,147	2,582 (353~8,198)	7,152	37,910	94.8%	55.1%
Normal -subject3	8,242	6,996	2,504 (326~8,892)	7,452	42,534	92.0%	55.7%
Normal -subject4	6,346	4,991	2,700 (320~8,416)	7,766	60,268	96.1%	55.7%
Normal -subject5	4,567	3,097	2,912 (298~9,285)	7,998	67,717	95.0%	49.0%
AEH-subject1	6,159	5,517	1,587 (329~8,602)	4,143	49,714	94.6%	58.6%
AEH-subject2	9,852	9,190	2,138 (149~8,262)	6,258	44,016	94.6%	56.2%
AEH-subject3	7,726	7,170	2,734 (316~7,626)	8,766	45,031	92.6%	63.3%
AEH-subject4	7,431	6,356	2,200 (215~8,233)	5,973	46,266	93.9%	65.0%
AEH-subject5	8,436	6,586	2,084 (348~8,905)	5,922	38,558	96.4%	59.8%
EEC-subject1	11,868	9,749	1,322 (255~7,332)	3,652	28,134	94.9%	59.7%
EEC-subject2	8,810	6,840	1,279 (321~7,794)	3,030	33,816	85.8%	52.9%
EEC-subject3	4,816	3,645	1,726 (260~9,222)	6,092	61,572	91.2%	60.2%
EEC-subject4	13,207	11,082	1,718 (278~7,716)	4,420	29,200	95.6%	67.2%
EEC-subject5	12,407	10,174	1,268 (340~7,403)	3,076	23,430	95.7%	64.5%

Reviewer #2 (Remarks to the Author):

The authors have done a great job with addressing my concerns in the revised manuscript.

In particular, they increased the sample size, updated analyses, and de-emphasized or toned down sections that may have had overreaching claims.

In the current form, I find the work to be useful for the community.

Response:

We greatly appreciate the reviewer's comments and suggestions about this work.

Reviewer #3 (Remarks to the Author):

The major update to the paper is the increase in sample size from 3 to 5. It is still a relatively small sample size, but I believe the manuscript is well written and care is taken not to over-interpret the results and as such believe that the authors have answered my queries. Some minor points mentioned below remain.

Major point 2: It is not clear from their response if they have stratified by subtype. The data from the TCGA analysis should not be used as a whole as it may include other subtypes that are non-endometrioid endometrial carcinoma e.g. Squamous cell carcinoma, small cell carcinoma and serous carcinoma. It should be confirmed that this has been checked and non-endometrioid cancers excluded from the analysis. If they are present they are likely to only be small numbers, but should still be excluded.

Response:

Thank you for your constructive suggestions. In this work, we analyzed data from endometrioid endometrial cancer only in TCGA database. To make it more clearly, we have rephrased the sentences. *“The TPM expression data were downloaded from Uterine Corpus Endometrial Carcinoma (TCGA-UCEC) database using the R package TCGAbiolinks (62). We included 411 samples with endometrioid endometrial cancer and 35 normal samples out of the 589 samples for downstream expression level comparison.”* (Lines 533-537)

Figure 3D- I may have misinterpreted, but the authors suggest that carcinogenesis is derived from unciliated glandular cells. I cannot see the unciliated glandular cells in this figure. Are these the same as the glandular cells?

Response:

Thank you very much for your comments. In Figure 3d, except for ciliated cells, other cells (including the glandular cell, the luminal cell, oncogenic subpopulation and others) are all unciliated cells. For clarity, we modify the figure caption in **Figure 3d** (previous Figure 3D).

In response to the question about other pathology. It is appreciated that it is very unlikely to have true “controls” and some sort of pathology will exist, although these confounding factors should be mentioned in the discussion. Even with 5 sample numbers, these are relatively small and given these pathologies are postulated to change endometrial cell composition and gene expression in the endometrium it could influence the results and should be mentioned in the discussion.

Response:

We thank the reviewer for the comments and add the description of limitations in the discussion section. *“It should be mentioned here that we are unlikely to achieve access to materials from the same patient to study the evolution of normal endometrium to AEH to EEC. Alternatively, we employ individual samples from different patients. However, the sample size of scRNA-seq in this study is still limited. Given the heterogeneity of endometrial cancer in terms of the tumor microenvironment composition and the underlying molecular alterations, the interpretation of the findings in this study*

cannot reflect the situation of all EEC patients. Additional scRNA-seq samples and further in vitro and in vivo validation experiments are needed.” (Lines 403-409)